# T cells modulate the microglial response to brain ischemia

Corinne Benakis[1]*, Alba Simats[1], Sophie Tritschler[2], Steffanie Heindl[1], Simon Besson-Girard[1], Gemma Llovera[1], Kelsey Pinkham[1], Anna Kolz[3], Alessio Ricci[1], Fabian J Theis[2], Stefan Bittner[4], Özgün Gökce[1,5], Anneli Peters[3,6], Arthur Liesz[1,5]*

[1]Institute for Stroke and Dementia Research, University Hospital, LMU Munich, Munich, Germany; [2]Institute of Diabetes and Regeneration Research, Institute of Computational Biology, Helmholtz Zentrum München, Neuherberg, Germany; [3]Institute of Clinical Neuroimmunology, University Hospital, LMU Munich, Munich, Germany; [4]Department of Neurology, Focus Program Translational Neuroscience (FTN) and Immunotherapy (FZI), RhineMain Neuroscience Network (rmn(2)), University Medical Center of the Johannes Gutenberg University Mainz, Mainz, Germany; [5]Munich Cluster for Systems Neurology (SyNergy), Munich, Germany; [6]Biomedical Center (BMC), Faculty of Medicine, LMU Munich, Munich, Germany

*For correspondence:
Corinne.Benakis@med.uni-muenchen.de (CB);
Arthur.Liesz@med.uni-muenchen.de (AL)

**Abstract** Neuroinflammation after stroke is characterized by the activation of resident microglia and the invasion of circulating leukocytes into the brain. Although lymphocytes infiltrate the brain in small number, they have been consistently demonstrated to be the most potent leukocyte subpopulation contributing to secondary inflammatory brain injury. However, the exact mechanism of how this minimal number of lymphocytes can profoundly affect stroke outcome is still largely elusive. Here, using a mouse model for ischemic stroke, we demonstrated that early activation of microglia in response to stroke is differentially regulated by distinct T cell subpopulations – with $T_{H1}$ cells inducing a type I INF signaling in microglia and regulatory T cells ($T_{REG}$) cells promoting microglial genes associated with chemotaxis. Acute treatment with engineered T cells overexpressing IL-10 administered into the cisterna magna after stroke induces a switch of microglial gene expression to a profile associated with pro-regenerative functions. Whereas microglia polarization by T cell subsets did not affect the acute development of the infarct volume, these findings substantiate the role of T cells in stroke by polarizing the microglial phenotype. Targeting T cell-microglia interactions can have direct translational relevance for further development of immune-targeted therapies for stroke and other neuroinflammatory conditions.

## Editor's evaluation

This manuscript should be of interest to neuroimmunologists investigating how microglia may be manipulated to improve neuroinflammation in stroke and beyond. The data support the hypothesis that manipulation of lymphocytes and the cytokines they secrete may be an effective therapeutic strategy to modulate inflammation and improve the outcome after stroke.

## Introduction

Among peripheral leukocytes invading the injured brain, T cells have been consistently identified as the invading leukocyte subpopulation with the largest impact on secondary neurodegeneration and modulation of the ischemic brain damage (*Kleinschnitz et al., 2010*; *Liesz et al., 2011*). T cell

subpopulations have the potential to play either a neuroprotective or a deleterious role in post-stroke neuroinflammation. In particular, the pro-inflammatory $T_{H1}$, $T_{H17}$ subsets of $T_{HELPER}$ cells, and IL-17-producing γδ T cells have been shown to induce secondary neurotoxicity, leading to infarct expansion with worse functional outcome (*Gelderblom et al., 2012*; *Shichita et al., 2009*), whereas regulatory T cells ($T_{REG}$) exert anti-inflammatory and neuroprotective function suppressing an excessive inflammatory reaction to the brain infarct. Recruitment of peripheral immune cells is not limited to the brain parenchyma since an accumulation of T cells is observed in the choroid plexus and the meninges after stroke (*Benakis et al., 2016*; *Llovera et al., 2017*). Attempt in blocking the recruitment of peripheral effector T cells diminished neuronal damage in different cerebral ischemic models, resulting in improvement of stroke outcome and suggesting a possible therapeutic target (*Liesz et al., 2011*; *Llovera et al., 2015*). Considering the relatively low number of only a few thousand lymphocytes invading the brain after stroke compared to more than 50 times higher cell count of innate immune cells (invading and resident) in the post-stroke brain (*Gelderblom et al., 2009*), it is surprising to observe such a dramatic effect of a small number of T cells on the neuroinflammatory response to stroke.

Therefore, we hypothesized that T cells have a polarizing effect on microglial function. In turn, microglia – as the most abundant immune cell population in the ischemic brain – could amplify the T cells' impact on the cerebral immune milieu. Indeed, microglia interact with T cells via either cell-to-cell contact, cytokine-mediated communication, or antigen presentation, leading to activation/polarization of adaptive immune cells entering the brain (*Goldmann and Prinz, 2013*). T cell-microglia interaction can further influence the neuroinflammatory response in experimental models of multiple sclerosis (*Dong and Yong, 2019*) and possibly in stroke (*Wang et al., 2016*). In fact, recent evidence suggests a crosstalk between microglia and T cells as a key determinant of neuronal plasticity during recovery from brain injury (*Shi et al., 2021*). However, while the influence of microglia/macrophages on T cells has been well studied, it is still unclear how in reverse the T cells influence microglial function, and whether early interaction of T cells with microglia in the acute response to stroke can have an immediate impact on microglia and further change the course of disease progression.

Using morphological analysis, single-cell sequencing and adoptive transfer models of ex vivo differentiated $T_{HELPER}$ cell subpopulations, we performed an in-depth analysis of the immunomodulatory effects of T cells on microglial polarization. Better understanding of the T cell-microglia crosstalk holds the potential to use polarized T cells as a therapeutic approach with large impact on the cerebral inflammatory milieu potentiated by resident microglia.

## Results

### Lymphocytes modulate the activation state of microglia in response to stroke

First, we investigated the effect of lymphocytes on microglial morphology and transcriptome in male *Rag1*$^{-/-}$ mice deficient in T and B lymphocytes after experimental stroke using the distal occlusion of the middle cerebral artery (dMCAO; *Llovera et al., 2014*; *Figure 1A*). Microglia were analyzed 5 days after stroke at the lesion border because this region and this acute time point were identified as the maximal cerebral leukocyte infiltration as previously published (*Llovera et al., 2017*) and exemplified in (*Figure 1B*). CD3+ T cells were localized at the infarct border as quantified on immunohistochemistry coronal sections and represented in the cumulative topographic maps 5 days after stroke (*Figure 1C*). Using an automated morphological analysis of IBA1 positive cells (*Heindl et al., 2018*) located at the perilesional (ipsilateral) cortex, where CD3+ T cells accumulated (*Figure 1B*), we identified that microglia of *Rag1*$^{-/-}$ mice displayed extended ramifications and a lower sphericity compared to microglia of wild-type (WT) mice, indicating a less reactive phenotype of microglia in the absence of lymphocytes (*Figure 1C, D*). In contrast, microglial morphology remained relatively unchanged between *Rag1*$^{-/-}$ and WT mice in the contralateral (unaffected) hemisphere which does not show recruitment of lymphocyte in considerable amounts, supporting the role of local lymphocyte infiltration for changing microglial morphology. Because the absence of lymphocytes prevents microglial morphological changes toward a reactive state, we asked whether the ischemic lesion is decreased in lymphocyte-deficient mice lacking microglia. We depleted microglia using PLX5622 incorporated in the mouse diet for 2 weeks prior inducing dMCAO (*Figure 1E, F*). Surprisingly, we

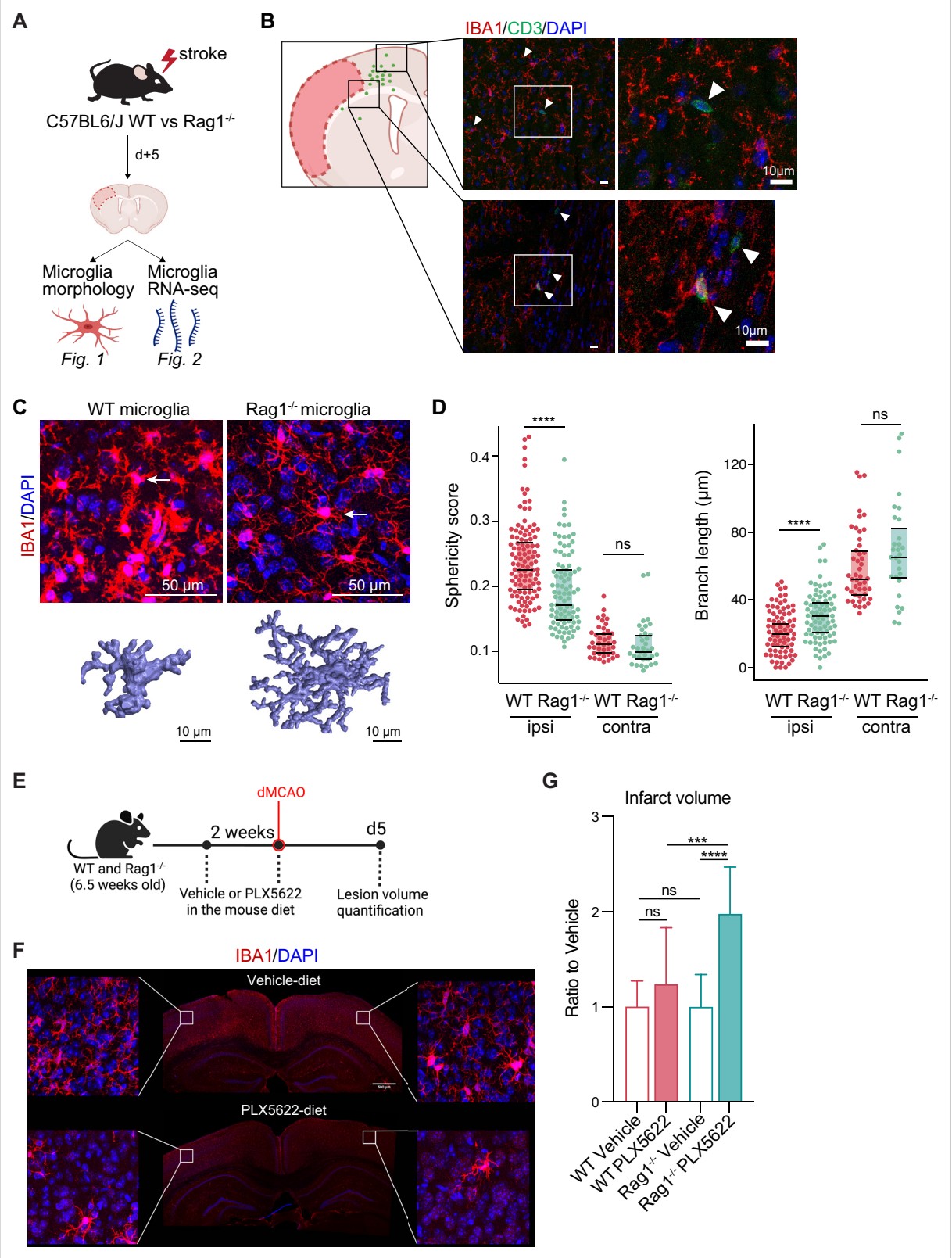

**Figure 1.** Lymphocytes influence microglia morphology after stroke. (**A**) Schematic of the experimental design: morphological analysis of microglia and transcriptomic profile of sorted microglia were performed in naïve mice or 5 days after stroke in wild-type (WT) and *Rag1⁻/⁻* mice. (**B**) Top left, cumulative topographic maps of CD3+ T cells 5 days after stroke. Cells were accumulated from one section at bregma level of five animals each. Each cell is represented as a single dot. The infarct is depicted in pink. Right, representative immunohistochemistry images of microglia (IBA1, red) and T cells (CD3,

*Figure 1 continued on next page*

*Figure 1 continued*

green) in the perilesional area 5 days after stroke in WT mice. 4',6-Diamidin-2-phenylindol (DAPI; blue) was used as nuclear dye. Bar scale indicates 10 µm. (**C**) Top, representative images of IBA1+ microglial cells in the perilesional region (900 µm distal to the infarct border, cortical layer 4). Bottom, three-dimensional (3D) reconstruction of microglia in WT and *Rag1⁻/⁻* mice. (**D**) Morphological analysis of microglia in the peri-infarct area (ipsi) and in the contralateral hemisphere (contra) for two representative features: sphericity and branch length (µm) in WT (red) and *Rag1⁻/⁻* (green) mice. Each dot corresponds to one microglial cell; n=3mice per condition; ns, non significant; ****, p<0.0001. Wilcoxon rank sum test with continuity correction and Bonferroni post-hoc correction for multiple testing. (**E**) WT and *Rag1⁻/⁻* mice were fed a chow diet containing the CSF1 receptor antagonist, PLX5622 (1200 ppm in mouse chow) for 2 weeks and until 5 days post-stroke to eliminate microglia from the brain, and another group of mice were fed with a control chow diet (vehicle). Infarct volumetry was quantified on cresyl violet staining 5 days after distal occlusion of the middle cerebral artery (dMCAO). (**F**) Images show that 2 weeks of PLX5622-diet almost completely depleted microglia from the brain (IBA1, red; 4',6-Diamidin-2-phenylindol [DAPI], blue), scale bar: 500 µm. (**G**) Infarct volumes of PLX5622-treated mice as a ratio to vehicle-treated mice 5 days after stroke; One-way ANOVA and Tukey's multiple comparison test; n=13–15 mice per condition; bar graphs show the mean and the standard deviation (SD); ns, non significant; ***, p<0.001; ****, p<0.0001.

found a significant increase of the infarct volume in *Rag1⁻/⁻* mice depleted of microglia in comparison to vehicle-treated *Rag1⁻/⁻* mice, whereas depletion of microglia in WT mice has no significant effect (**Figure 1G**), showing that depletion of microglia in WT mice does not have the same effect on stroke as in the lymphocyte deficient mice. These data suggest that microglia-T cell interaction is required to influence the development of the infarct. To better understand this interaction, we investigated the functional implications of cerebral lymphocyte invasion for microglia by single-cell sequencing (10× Genomics pipeline) of sorted CD45⁺CD11b⁺ cells.

CD45⁺CD11b⁺ myeloid cells were sorted by flow cytometry from naïve mice or 5 days after stroke (pool of 3 mice per condition; **Figure 1A** and **Figure 2A**). To better discriminate the transcriptional signature of microglial cells from other CD45⁺CD11b⁺ myeloid cells, we performed an unsupervised clustering analysis and identified 15 distinct clusters across conditions (**Figure 2—figure supplement 1A**). Based on the expression of previously defined markers of homeostatic and reactive microglia per cell cluster (high gene expression of *Fcrls*, *P2ry12*, and *Trem2*; low expression of *Itgax*, *Ccr2*, and *Lyz2*; **Keren-Shaul et al., 2017**; **Miron and Priller, 2020**; **Prinz and Priller, 2014**), five clusters were annotated as microglial cells (**Figure 2B**). We then performed a subclustering analysis on only the microglia cells (**Figure 2—figure supplement 1B**) and identified subpopulations showing either a transcriptomic profile preferentially associated with homeostatic microglial function (clusters 0, 1, 4, and 6) or a profile of reactive microglia (clusters 2, 3, 5, and 7; **Figure 2C**, left plot and **Figure 2—figure supplement 1C**). The cell distribution across condition highlighted that stroke is the main driver of the microglial transcriptomic changes, both in WT and *Rag1⁻/⁻* mice (**Figure 2C**, right plot). Volcano plots of the differentially expressed genes revealed that *Apoe* and *Cd74* were down-regulated in naïve *Rag1⁻/⁻* mice, indicating that the transcriptional profile of microglia is affected by the absence of lymphocytes in homeostatic condition (**Figure 2D**, left plot). Interestingly, several genes, known to define the signature of the disease-associated microglia (DAMs, Keren-Shaul et al., 2017), were up-regulated in *Rag1⁻/⁻* mice after stroke (*Apoe*, *Cd74*, *Cstb*, *Lgals3*, and *Lyz2*; **Figure 2D**, right plot). When we compared the stroke-associated microglial genes between WT and *Rag1⁻/⁻* mice, we found that 67 genes – including the majority of the DAM signature (*Apoe*, *B2m*, *Cstb*, *Lgals3*, *Lyz2*, and *Spp1*) – were not specific to the lymphocyte deficiency (**Figure 2E**), whereas 120 stroke-associated microglial genes were only present in mice lacking lymphocytes, such as genes involved in cytokine signaling and chemotaxis (i.e. *Cd74*, *Ccl2*, *Ccl7*, *H2-Ab1*, *Infgr1*, *Mif*, *Pf4*, and *Tnf*).

The microglial reaction to stroke causes a gradual shift from the homeostatic transcriptomic profile to a reactive state. In order to capture differences in the microglia transcriptome along its transition phase, we performed single-cell trajectory inference analysis (**Figure 2—figure supplement 2**). Partition-based graph abstraction (PAGA) revealed two distinct paths with high connectivity from the homeostatic (naïve) microglia cluster (root cluster) to the reactive (stroke) microglia cluster (end cluster; **Figure 2—figure supplement 2A**). Interestingly, the number of microglial cells in stroke *Rag1⁻/⁻* increased in the end cluster and was decreased along the trajectory path 2 in comparison to stroke WT (**Figure 2—figure supplement 2B**), suggesting that lymphocytes influence the transition of a microglia subpopulation from the homoeostatic to the reactive state. Differential gene expression analysis between the root and end clusters of the trajectory path 2 in WT and *Rag1⁻/⁻* mice (**Figure 2—figure supplement 2C**) revealed that genes associated with ribosomal metabolic processes and mitochondrial ribosomal proteins were specifically enriched, whereas genes associated with phagocytosis

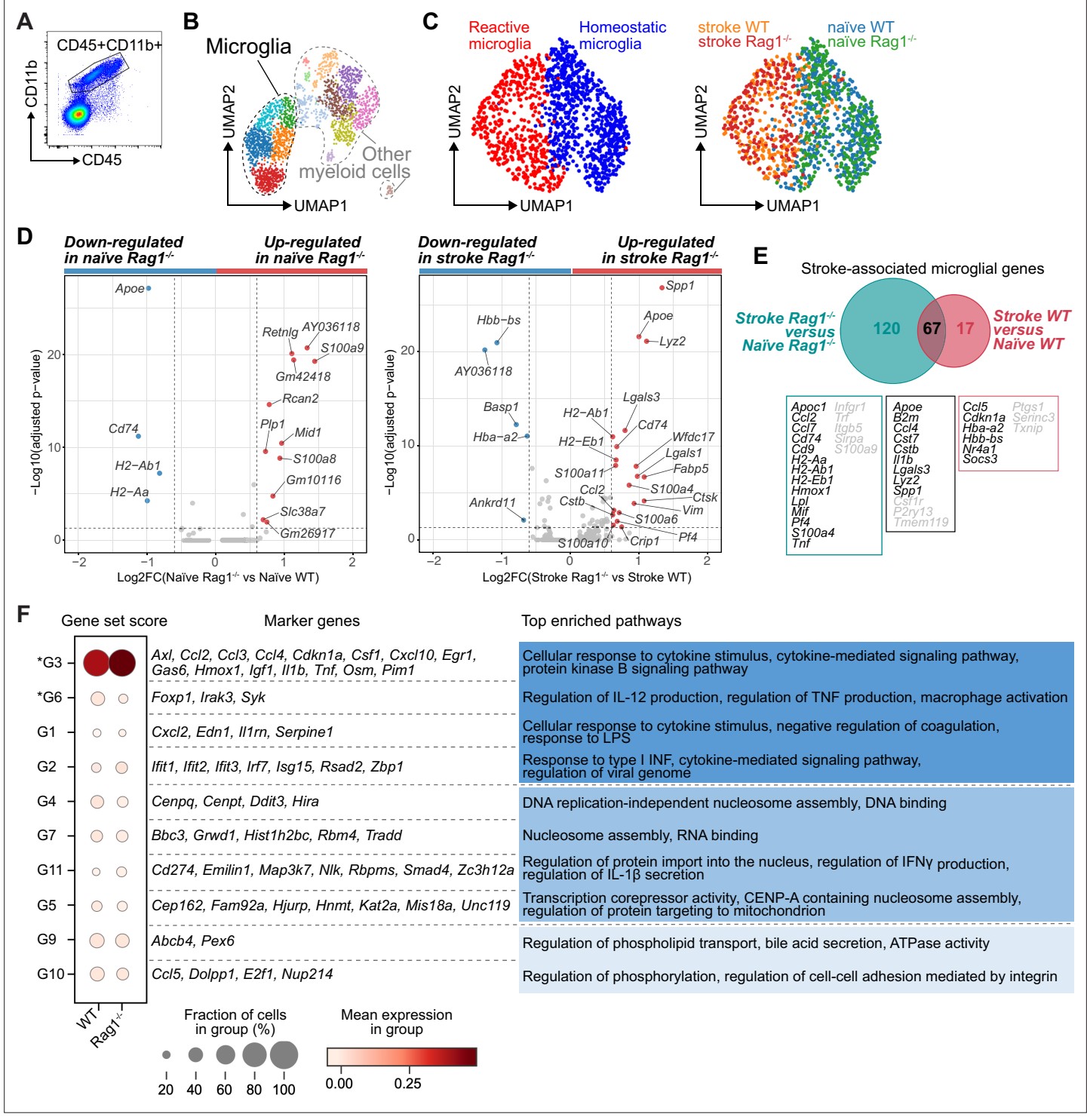

**Figure 2.** Lymphocytes influence microglia transcriptional signature. (**A**) CD45+CD11b+ cells were sorted from the ipsilateral hemisphere in naïve mice or 5 days after stroke in wild-type (WT) and *Rag1*−/− (3 mice per condition), and RNA was isolated for single cell RNA sequencing (10× Genomics). (**B**) Uniform manifold approximation and projection 2D space (UMAP) plots of 2345 CD45+CD11b+ cells colored by 15 distinct transcriptional clusters (*Figure 2—figure supplement 1A*). (**C**) Clustering of the microglia subset color-coded by homeostatic and reactive microglia (right) and by conditions (left). (**D**) Volcano plots of the differentially expressed genes in microglia in naïve and stroke condition. Dotted lines indicate an adjusted p-value≤0.05 and FC = 1.5. (**E**) Number of microglial genes regulated after stroke in comparison to naïve condition in *Rag1*−/− and WT mice. 67 genes were common to both genotypes, 17 genes were specifically regulated in WT mice and 120 genes only in *Rag1*−/− mice. Boxes indicate key microglial genes in each condition (genes indicated in gray were down-regulated after stroke). (**F**) Selected gene sets of highly correlated and anti-correlated genes

*Figure 2 continued on next page*

*Figure 2 continued*

based on trajectory inference analysis in stroke condition (***Figure 2—figure supplement 2E–G***). Mean gene set activation score in WT and *Rag1*⁻/⁻ cells, selected marker genes, and top enriched gene ontology pathways associated to each gene set. Gene sets were classified by p-value (the lowest p-value at the top, asterisks [*] indicate significant difference between genotype in stroke condition) and by similar pathways, such as: pathways related to inflammation (dark blue), pathways related to DNA/RNA regulation (blue), and lipid pathways (light blue).

The online version of this article includes the following figure supplement(s) for figure 2:

**Figure supplement 1.** Transcriptomic analysis of microglia isolated from wild-type (WT) and *Rag1*⁻/⁻ mice in naïve and stroke conditions.

**Figure supplement 2.** Microglia single cell trajectory inference in wild-type (WT) and *Rag1*⁻/⁻ mice in naïve and stroke conditions.

**Figure supplement 3.** Immune cell infiltration in wild-type (WT) and *Rag1*⁻/⁻ mice after stroke.

were down-regulated in microglia of *Rag1*⁻/⁻ mice (***Figure 2—figure supplement 2D***). We then clustered genes into groups of correlating and anti-correlating genes and investigated the activation of these gene sets along the identified trajectory path 2 in stroke condition only (***Figure 2—figure supplement 2E–G***). Gene sets which were significantly different between WT and *Rag1*⁻/⁻ mice after stroke revealed that the absence of lymphocytes significantly reduces microglial genes associated with macrophage activation state (G6: *Foxp1*, *Syk Shi et al., 2008*; *Tabata et al., 2020*), whereas genes associated with cytokine/chemokine stimulus were enriched (G3: *Il1b*, *Tnf*, *Csf1*, and *Ccl2*) in *Rag1*⁻/⁻ in comparison to WT microglia (***Figure 2F***). These results revealed that stroke is the primary driver of transcriptomic changes in microglia at this acute time point and that lymphocytes modulate the activation status of a subset of stroke-associated microglial cells associated with cytokine/chemokine regulation in the post-ischemic brain.

Because *Rag1*⁻/⁻ mice lack mature T cells and B cells, it is possible that the observed morphological and transcriptional changes of microglia may be due to B cells or other myeloid cell types. We performed flow cytometry analysis of the ipsilesional hemisphere 5 days after dMCAO in WT and *Rag1*⁻/⁻ mice. First, we demonstrate that T cells are 14 times more abundant than B cells in the ipsilesional hemisphere in WT mice (***Figure 2—figure supplement 3A***). In addition, the abundance of myeloid cell subsets is not affected by the Rag1 gene deletion (***Figure 2—figure supplement 3B***). These data support the hypothesis that lymphocytes and most likely T cells are the main contributor to the observed microglial phenotype at this time point after stroke. Because previous findings showed the CD4+ T cell subpopulations exert distinct effects during the post-stroke immune response (***Liesz et al., 2009***; ***Iliff et al., 2012***), we next determined whether functionally different T cell subsets induce or suppress genes in microglia related to cytokine production or cell migration as observed in ***Figure 2E, F***.

## T_HELPER_ cell subpopulations drive the distinct polarization of microglia

To test whether microglial phenotypes can be specifically skewed by the CD4+ T cell subsets of functionally opposing T_HELPER_ cell subpopulations, we differentiated T_H1_ and T_REG_ in vitro (***Figure 3—figure supplement 1A***) and tested whether these T_HELPER_ cells can reprogram the stroke-associated microglia. Differentiated T cells or vehicle were injected into the cisterna magna (CM) of lymphocyte-deficient *Rag1*⁻/⁻ mice 24 hr after stroke. Microglia cells CD45+CD11b+ were sorted from the ipsilesional hemisphere 24 hr after polarized T_HELPER_ cell (T_H1_ or T_REG_ cells) or vehicle administration (***Figure 3A***). The transcriptional profile of microglia induced by T_REG_ cells was more similar to vehicle treated *Rag1*⁻/⁻ mice (named control [CT]) than microglial gene expression induced by T_H1_, as shown in the heatmap and volcano plots of the differentially expressed genes (p<0.05 and |fold change|>1.5) with 34 and 12 microglial genes regulated in T_H1_ or T_REG_ conditions compared to control injection, respectively (***Figure 3B, C***). Gene ontology analysis of the differentially up-regulated genes revealed T_H1_-dependent pathways associated with antigen presentation, response to cytokines, and regulation of type I INF, whereas T_REG_-dependent microglial genes were associated with chemotaxis (***Figure 3D***). These results demonstrate the potency of T cell subpopulations to differentially skew the microglial transcriptome toward distinct phenotypes previously associated with different cellular functions. In particular, we found that T_H1_ polarized microglia toward an antigen-immunocompetent phenotype (*Cd74* and *Lag3*) and expression of INF response-related genes (*Irf7* and *Stat1*). This profile of microglial response was previously associated with a pronounced immune response during the later stages of neurodegeneration (***Mathys et al., 2017***). In addition, after experimental stroke, the T_H1_-mediated effects on the

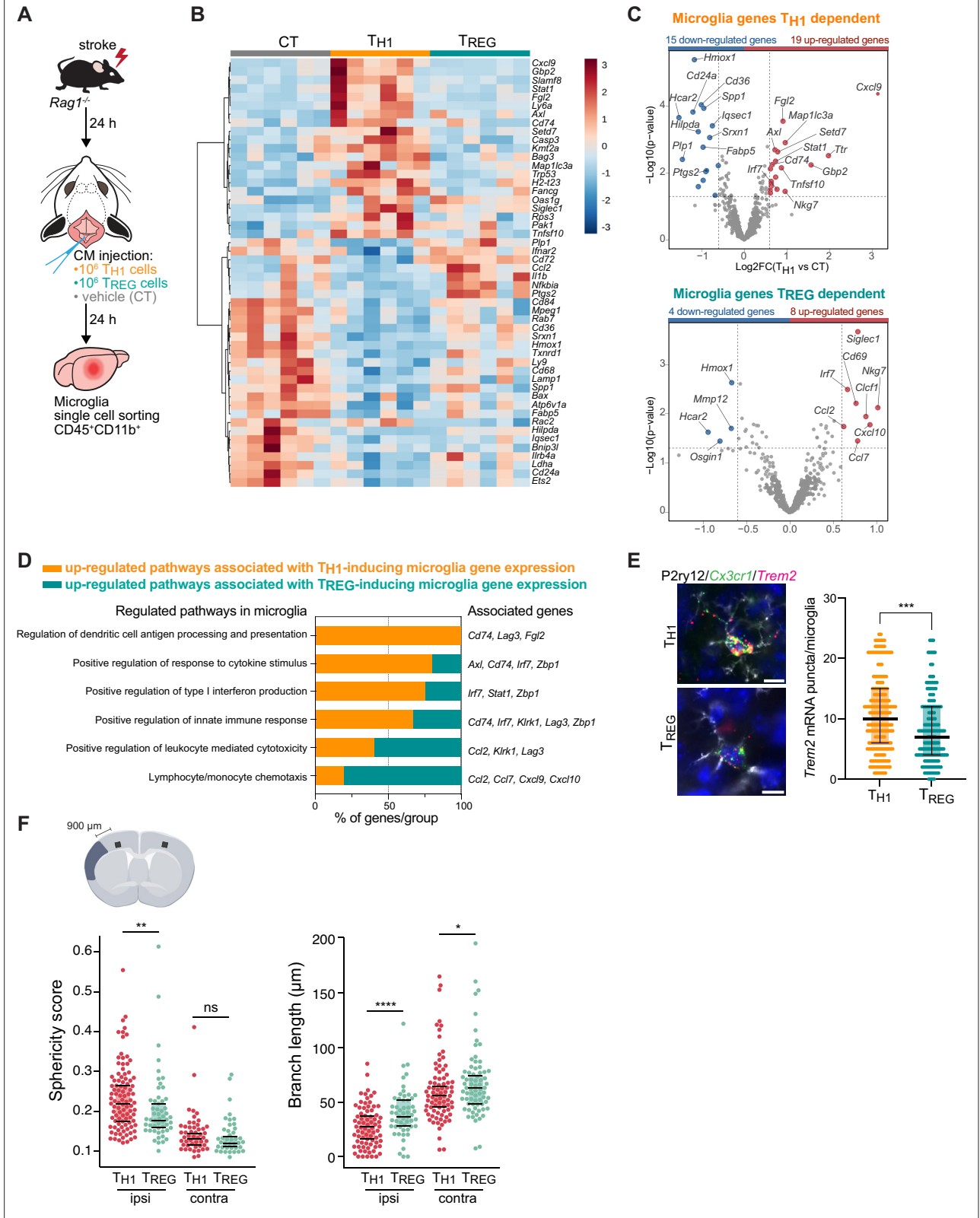

**Figure 3.** T$_{H1}$ and regulatory T cells (T$_{REG}$) cells influence microglia gene expression after stroke. (**A**) Naïve CD4 cells were polarized in vitro to T$_{H1}$ or T$_{REG}$ phenotype (*Figure 3—figure supplement 1A*). One million cells (T$_{H1}$ or T$_{REG}$ cells) or vehicle (control, CT) were injected into the cisterna magna (CM) in *Rag1*$^{−/−}$ mice 24 hr after stroke induction (n=6 mice per condition). Microglia cells CD45+CD11b+ were sorted from the ipsilesional hemisphere, and RNA was extracted. Gene expression analysis was performed using the Neuroinflammation Panel profiling kit on the Nanostring platform. In a second

*Figure 3 continued on next page*

*Figure 3 continued*

set of experiment, 100 μm coronal sections were proceeded for single-molecule fluorescence in situ hybridization (smFISH) or microglia morphology. (**B**) Heatmap representation of microglia gene expression between conditions: control (CT; vehicle administration of PBS), $T_{H1}$ or $T_{REG}$. (**C**) Up- and down-regulated differentially expressed genes between either isolated microglia from $T_{H1}$- (top) and $T_{REG}$- (bottom) treated $Rag1^{-/-}$ mice relative to control condition (microglia isolated from $Rag1^{-/-}$ mice treated with vehicle, genes are color-coded accordingly to a p-value<0.05 and |fold change|>1.5). (**D**) Pathway analysis was performed for the up-regulated genes in each condition using the ClueGO package from Cytoscape. (**E**) Higher amount of *Trem2* mRNA puncta (red) per *Cx3cr1*-positive (green) in P2ry12-labeled microglia (white) in $T_{H1}$-treated mice in comparison the $T_{REG}$-treated mice. 4′,6-Diamidin-2-phenylindol (DAPI; blue) was used as nuclear dye. Scale bar = 10 μm. Each dot corresponds to one P2ry12-microglial cell; n=3 mice per condition; graphs show the median with interquartile range . (**F**) Morphological analysis of IBA1+ microglia in the ipsilateral (900 μm distal to the infarct border, cortical layer 4) and contralateral hemisphere, as shown in the representative coronal section. Sphericity score and branch length (μm) of microglia treated with $T_{H1}$ (orange) or $T_{REG}$ cells (green). Each dot corresponds to one microglial cells; ns, non significant; n=3 mice per condition. *, p<0.05; **, p<0.01; ****, p<0.0001. Wilcoxon rank sum test with continuity correction and Bonferroni post-hoc correction for multiple testing.

The online version of this article includes the following figure supplement(s) for figure 3:

**Figure supplement 1.** T cell polarization in vitro and infarct volumetry in $Rag1^{-/-}$.

microglial transcriptomic profile were associated with an increase of *Trem2* expression, a key marker of DAM in various brain disorders, in comparison to microglia primed by $T_{REG}$ cells (***Figure 3E***). In contrast, $T_{REG}$ cells promoted the expression of chemokines/cytokines in microglia (*Ccl2*, *Ccl7*, and *Cxcl10*), which can have either pro-regenerative or detrimental effects such as the regulation of leukocyte chemotaxis to the injured brain (***Llovera et al., 2017***), mechanisms of protective preconditioning (***Garcia-Bonilla et al., 2014***) or promoting neuronal stem cell recruitment and angiogenesis (***Andres et al., 2011***; ***Lee et al., 2012***; ***Liu et al., 2007***). Interestingly, this set of chemotactic genes induced by $T_{REG}$ cells were also differentially expressed in microglia isolated from $Rag1^{-/-}$ mice in comparison to WT mice (***Figure 2E, F***), suggesting that the stroke associated microglia may lose their chemotactic properties in a T cell-dependent manner and especially a $T_{REG}$ cell dependent (***Chen and Bromberg, 2006***). Because $T_{REG}$ cells induce beneficial functions in cerebral ischemia (***Liesz et al., 2009***), it could be speculated that restoring microglial chemotactic behavior by $T_{REG}$ cells could contribute to recovery. These transcriptomic differences in microglia related to the in vivo $T_{H1}$ or $T_{REG}$ cell exposure were also reflected by the difference in the morphology of microglia between these conditions. Microglia displayed a reactive state as shown by a more spherical and less branched morphology in $T_{H1}$ cell-injected compared to $T_{REG}$-injected mice (***Figure 3F***) similarly to microglia in $Rag1^{-/-}$ mice (***Figure 1C, D***). Interestingly, these morphological changes were not only restricted to the ipsilesional hemisphere as seen in $Rag1^{-/-}$ not reconstituted with T cell subsets (***Figure 1D***) but were also observed in the contralateral hemisphere, suggesting possible brain-wide effects of differentiated $T_{HELPER}$ cells injected to the CSF compartment. In accordance, we found that intra-CM injection of eGFP-labeled $T_{H1}$ cells to $Rag1^{-/-}$ mice after stroke was primarily recruited to the ischemic brain parenchyma but was additionally localized in border tissues including the meninges, and some CM-injected cells even circulated and could be detected in the spleen ( ***Figure 4A, B*** and ***Figure 4—figure supplement 1***). Importantly, no difference in infarct volumes was observed between WT and $Rag1^{-/-}$ mice (***Figure 3—figure supplement 1B***) and in $T_{H1}$ or $T_{REG}$-supplemented $Rag1^{-/-}$ mice (***Figure 3—figure supplement 1C***), suggesting the transcriptional changes observed in microglia are primarily due to T cell subsets and not biased by differences in stroke severity. Together, these findings support that polarized T cells are recruited to the infarction site and may modify in situ the inflammatory micromilieu.

## Engineered T cells overexpressing IL-10 induce a pro-regenerative transcriptomic profile in microglia

In order to further explore the implication of $T_{REG}$-microglia interactions to modulate the post-stroke inflammatory environment, we tested the therapeutic potential of the known anti-inflammatory properties of $T_{REG}$ via IL10 on the local microglial immune milieu. We engineered T cells by viral transfection to overexpress the anti-inflammatory cytokine IL-10 (eTc-IL10; ***Figure 4—figure supplement 1B, C***). In a therapeutic approach, we injected eTc-IL10 cells into the CM of WT mice 4 hr after stroke – a translationally relevant time window considering a similar time window for acute therapy with thrombolytics in stroke patients (***Figure 4C***). We investigated whether eTc-IL10 treatment affected stroke outcome but did not find any difference in infarct volumes between conditions (***Figure 4D***). This is in accordance with the concept of early ischemic lesion formation in stroke which is not being affected

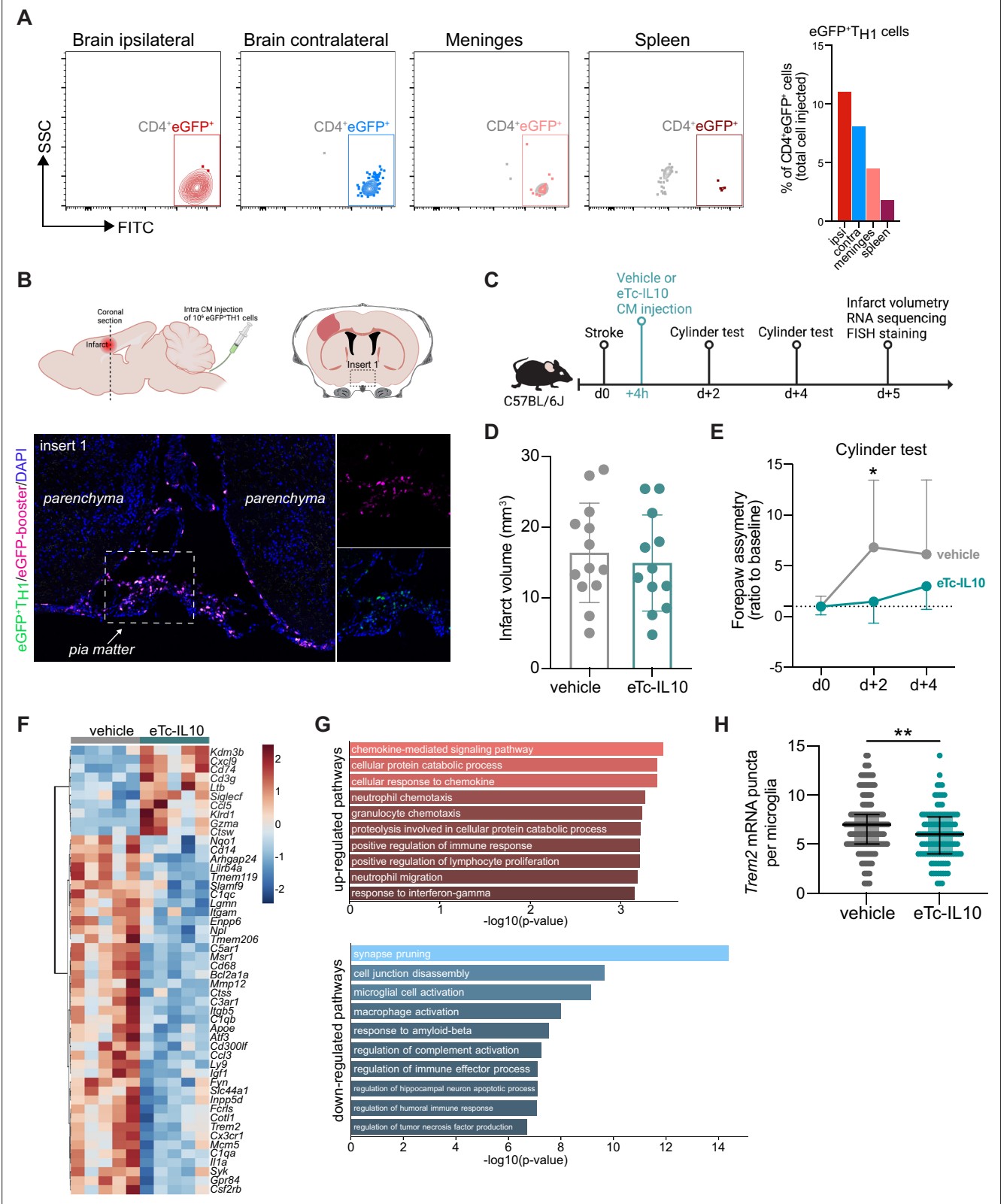

**Figure 4.** Acute post-stroke treatment with engineered T cells overexpressing IL-10 modulates microglial activation and ameliorates functional deficit. (**A and B**) Flow cytometry analysis and whole skull-brain coronal sections of $10^6$ eGFP+T$_{H1}$ cells injected into the cisterna magna (CM) of *Rag1*$^{-/-}$ mice 24 hr after stroke. Samples were collected 4 hr after CM injection for further analysis. (**A**) Flow cytometry plots showing CD4+eGFP+ cells isolated from the brain (ipsilateral and contralateral hemispheres), meninges, and spleen (the detailed gating strategy is shown in ***Figure 4—figure supplement 1A***).

*Figure 4 continued on next page*

*Figure 4 continued*

The graph represents the percentage of eGFP+T$_{H1}$ cells relative to the total number of cells injected in the CM ($10^6$ eGFP+T$_{H1}$ cells). (**B**) Coronal section showing eGFP+T$_{H1}$ cells in the meninges. Insert 1 indicates a representative photomicrograph of eGFP+T$_{H1}$ cells counterstained with an eGFP-booster (magenta), and cell nuclei are stained with 4',6-Diamidin-2-phenylindol (DAPI; blue). The magnified images of white boxed area show eGFP+T$_{H1}$ cells injected into the CM are located in the meninges. (**C**) Timeline of the experimental design. (**D**) Infarct volumes at 5 days after stroke in wild-type (WT) C57BL/6 J mice treated by CM administration of either T cells secreting IL-10 (eTc-IL10, $10^6$ naïve CD4+) cells transfected with a plasmid overexpressing IL-10, *Figure 4—figure supplement 1B, C, d* or vehicle (aCSF) 4 hr after stroke induction. (**E**) Percentage of assymetry in independent forepaw use ('0%' indicates symmetry) in mice treated with vehicle or eTc-IL10; *, p<0.05, ANOVA with Šídák's multiple comparisons test; n=12–13 mice per condition. (**F**) Heatmap representation of ipsilateral brain gene expression between vehicle and eTc-IL10 treated mice 5 days after stroke; n=5 mice per condition; one sample per condition was excluded due to unsatisfactory quality control check. (**G**) Selected gene ontology annotations for the 50 genes that were up-(top) and down-regulated (bottom) in the whole ipsilateral brain tissue of eTc-IL10 treated mice in comparison to vehicle treated mice. (**H**) Single-molecule fluorescence in situ hybridization (smFISH) analysis of brains from eTc-IL10 treated mice showed a reduction of *Trem2* mRNA puncta per *Cx3cr1*-positive microglia in the peri-infarct region in comparison to vehicle treated mice; **, p<0.01, Mann-Whitney U test; each dot corresponds to one microglial cell; n=3 mice per condition; graphs show the median with interquartile range.

The online version of this article includes the following figure supplement(s) for figure 4:

**Figure supplement 1.** Localization of polarized T cell and IL-10 plasmid construct.

by the delayed immunological mechanisms (*Dirnagl et al., 1999*). In contrast, mice receiving eTc-IL10 injection in the CM had a significant improvement of functional outcome at 48 hr after stroke as shown by a reduced forelimb asymmetry in comparison to vehicle-treated mice (*Figure 4E*). This might reflect the implication of inflammatory pathways and specifically cytokine secretion on functional deficits and delayed recovery after stroke in contrast to the early primary lesion development (*Filiano et al., 2017*; *Roth et al., 2020*). We then evaluated whether gene expression was altered after stroke upon eTc-IL10 treatment. RNA was isolated from the whole ischemic hemisphere, and neuroinflammatory genes were quantified using the Nanostring platform. Interestingly, we found that several genes associated with a DAM profile were down-regulated in mice treated with eTc-IL10 such as *Cd68*, *Apoe*, *Trem2*, *Tyrobp*, and *Cst7* (*Figure 4F* and *Figure 4—figure supplement 1D*). Gene ontology analysis revealed that T cell-derived IL-10 overexpression increased pathways associated with chemokine responses – similarly to *Rag1$^{-/-}$* mice reconstituted with T$_{REG}$ cells – and the down-regulation of several microglial effector functions such as spine pruning, phagocytosis, and complement activation (*Figure 4G*). Although the observed regulated genes are well known to be associated with microglial function, it is conceivable that in this analysis, other cell types than microglia, including various brain-invading myeloid cell subsets, could account for this effect since the whole ischemic brain tissue was processed for Nanostring analysis. This anti-inflammatory effect of eTc-IL10 treatment on microglia was confirmed by a reduction of *Trem2* mRNA in *Cx3cr1*+microglia from eTc-IL10 compared to vehicle-treated mice (*Figure 4H*). Since we observed a down regulation of genes associated with synapse pruning (*C1qa*, *C1qb*, and *C1qc*), microglia activation, and phagocytosis (*Apoe*, *Ctss*, *Trem2*, and *Cd68*) in mice treated with eTc-IL10, we postulate that acute intra-CM administration of eTc-IL10 induces a switch of the microglia gene signature possibly involved in promoting post-stroke recovery mechanisms.

## Discussion

The cellular constituents of the acute neuroinflammatory response to stroke have been well characterized, including microglial activation, leukocyte invasion, and the contribution of different lymphocyte subpopulation (*Anrather and Iadecola, 2016*). However, the reciprocal interactions of these different immune cell populations remain largely under-investigated in the context of brain injury. A better understanding of the T cell-polarizing effect on microglial function has strong translational implication since T cells may act as a 'Trojan horse' with large impact on the cerebral inflammatory milieu potentiated by microglia (*Cramer et al., 2018*).

Here, we established a mechanistic link between T cells and microglial morphology and transcriptomic signature in the context of stroke. We showed the distinct role of T cell subpopulations on switching microglial polarization state in response to stroke. Our results from transcriptomic analysis suggest that the microglia-polarizing effect of different T$_{HELPER}$ cell subpopulations is mainly mediated via their specific cytokine/chemokine secretion pattern. Microglia that were challenged with T$_{H1}$ cells expressed an up-regulation of genes associated with type I INF signaling – the key cytokine

secreted by the T$_{H1}$ subpopulation. In contrast, T$_{REG}$ cells modulated a gene set in microglia associated with chemotaxis-mediated mechanisms (*Ccl2*, *Ccl7*, and *Cxcl10*). Although it remains to be defined whether microglia primed by T$_{REG}$ cells contribute to the recruitment of other immune cells, especially of T$_{REG}$ cells inducing neuroprotective mechanisms. We also observed that T$_{REG}$ cells mediate a down-regulation of markers associated with reactive microglia such as the expression of *Trem2*, which have previously been described to be regulated by the T$_{REG}$-cytokine IL-10 (*Shemer et al., 2020*). These previous and our own results here clearly show the direct role of IL-10 in modulating microglial function. Likewise, using adult human microglial cells co-culture with T lymphocytes, others demonstrated an enrichment of IL-10 secretion upon direct cell-cell contact (*Chabot et al., 1999*). In addition, we previously reported using whole genome sequencing that intracerebroventricular injection of IL-10 is sufficient to modulate the neuroinflammatory response after experimental stroke (*Liesz et al., 2014*). However, we cannot exclude in this study the contribution of IL-10 from other lymphocyte subpopulations, particularly IL-10-producing regulatory B cells (*Bodhankar et al., 2013*; *Ortega et al., 2020*; *Seifert et al., 2018*), as we did not specifically deplete IL-10 in T cells.

An important caveat and potential key reason for the so far still pending success in harnessing the therapeutic function of IL-10 are its short half-life (less than 1 hr) and limited bioactivity after in vivo administration as a recombinant protein (*Le et al., 1997*; *Saxena et al., 2015*). Moreover, the systemic IL-10 application can have considerable and unforeseen side-effects due to the potentially divergent function of IL-10 on inflamed and homeostatic tissue, including direct effects on neurons, astrocytes, endothelial cells, and other cellular constituents of physiological brain function (*Saraiva et al., 2019*). Therefore, we aimed to take a different approach for the localized and sustained production of IL-10 at the inflamed peri-lesional brain parenchyma. For this, we took advantage of the potent capability of T cells to be specifically recruited and accumulated to the ischemic lesion site in order to deliver IL-10 from genetically engineered IL-10-overexpressing T cells (*Heindl et al., 2021*; *Llovera et al., 2017*). We demonstrated that IL-10 overexpression by this approach substantially modulated microglia gene expression by down-regulation of microglial gene signature associated with phagocytosis of synapses correlating with functional recovery after stroke. Interestingly, eTc-IL10 cells did not exclusively invade the injured brain but were also located in the meningeal compartment and could additionally contribute to functional recovery by resolving inflammation at these border structures or providing IL-10 to the brain parenchyma along CSF flow. This concept is in accordance with previous observations of meningeal immune cell accumulation after stroke (*Benakis et al., 2018*) and that meningeal T cell-derived cytokines may enter the brain via CSF flow and paravascular spaces (*Iliff et al., 2012*).

An important finding in this study was the observation that IL-10 overexpression in T cells modulated microglial genes involved in the complement pathway, phagocytosis, and synaptic pruning and was associated with a better functional outcome after stroke. Complement factors are localized to developing CNS synapses during periods of active synapse elimination and are required for normal brain wiring (*Schafer et al., 2012*). Inactive synapses tagged with complement proteins such as C1q may be eliminated by microglial cells. Likewise in the mature brain, early synapse loss is a hallmark of several neurodegenerative diseases (*Stephan et al., 2012*). Indeed, complement proteins are profoundly up-regulated in many CNS diseases prior to signs of neuron loss, suggesting mechanisms of complement-mediated synapse elimination regulated by microglia potentially driving disease progression (*Stephan et al., 2012*) and stroke recovery. It is therefore conceivable that T cells over-expressing IL-10 down-regulate the complement system in microglia and prevent excessive elimination of synapse and consequently protect against neuronal dysfunction. This is particularly of interest because microglia effector function has not only been associated with inflammatory neurodegenerative processes but recently also been shown to be neuroprotective (*Szalay et al., 2016*) by tightly monitoring neuronal status through somatic junctions (*Cserép et al., 2020*). Microglia interact with the extra-neuronal space by not only regulating the elimination of existing synapses but also by modifying the extracellular matrix to enable efficient synaptic remodeling (*Zaki and Cai, 2020*). Accordingly, we found T cell-dependent regulation of several microglial genes that can mediate such extracellular matrix modifications involved in phagocytosis and proteases (*Clstn1* and *Mmp12*, cathepsins and MMPs, respectively).

Whereas at this acute time point, the transcriptomic changes in microglia are mainly attributed to their reactivity to the tissue injury itself, we have been able to demonstrate that brain-invading T cells can specifically 'fine-tune' the transition of the stroke-associated microglia to a distinct cell morphology

and transcriptomic profile. Our data suggested that the anti-inflammatory $T_{REG}$ cells induce a shift of microglial genes associated with a homeostatic state and immune cell recruitment. However, the specific functional change of microglia induced by T cell subsets and biological significance for stroke remain to be further investigated. We postulate that the development of engineered T cells could have important translational implication by targeting a specific effector function of microglia with a relevant impact on the chronic progression of stroke pathobiology.

## Materials and methods

### Animal experiments

All animal procedures were performed in accordance with the guidelines for the use of experimental animals and were approved by the respective governmental committees (Licenses: 02-21-46 and 02-21-95; Regierungspraesidium Oberbayern, the Rhineland Palatinate Landesuntersuchungsamt Koblenz). Male WT C57BL6/J mice were purchased from Charles River, $Rag1^{-/-}$ mice (NOD.129S7[B6]-Rag-1tm1Mom/J) and eGFP-reporter mice (C57BL/6-Tg[CAG-EGFP]131Osb/LeySopJ) were bred and housed at the animal core facility of the Center for Stroke and Dementia Research (Munich, Germany). All mice were housed with free access to food and water at a 12 hr dark-light cycle. Data were excluded from all mice that died during surgery. Animals were randomly assigned to treatment groups, and all analyses were performed by investigators blinded to group allocation. All animal experiments were performed and reported in accordance with the ARRIVE guidelines (*Kilkenny et al., 2011*).

### Permanent distal middle cerebral artery occlusion model

Permanent coagulation of the middle cerebral artery (MCA) was performed as previously described (*Llovera et al., 2014*). Briefly, animals (male; age = 8–12 weeks) were anesthetized with volatile anesthesia (isoflurane in 30%$O_2$/70%$N_2O$) and placed in lateral position. After a skin incision between eye and ear, the temporal muscle was removed, and the MCA was identified. Then, a burr hole was drilled over the MCA, and the dura mater was removed. The MCA was permanently occluded using bipolar electrocoagulation forceps. Permanent occlusion of the MCA was visually verified before suturing the wound. During the surgery, body temperature was maintained using a feedback-controlled heating pad. Mice that developed a subarachnoid hemorrhage during surgery were excluded from the analysis.

### Cylinder test

To evaluate forepaw use and asymmetry, the cylinder test was performed 2 days prior to stroke (baseline) and day 2 and day 4 post stroke. Mice were placed in a transparent acrylic glass cylinder (diameter 8 cm; height: 25 cm) in front of two mirrors and videotaped. To assess independent forelimb use, contact with one forelimb (left and right forelimbs) during full rearing and landing performance of mice was scored by frame-to-frame analysis of recorded videos. Mice with forepaw preference at baseline (absolute value difference between right and left forepaws >10) were excluded from the analysis. All rearing movements during the trial were counted and used as indication of the animal's overall activity.

### Intra-CM injection

Mice were anesthetized with isoflurane in 30%$O_2$/70%$N_2O$ and fixed in a stereotaxic frame by the zygomatic arch, with the head slightly tilted to form an angle of 120° in relation to the body. A small incision was made at the nape of the neck between the ears to expose the neck muscles, which were bluntly dissected to expose the CM. Cannulas composed of a glass capillary (ID, inner diameter 0.67 mm; OD, outside diameter, 1.20 mm) attached to a polyethylene tubing (ID 0.86 mm and OD 1.52 mm; Fisher Scientific UK Ltd.) were used to perform the CM injections. Glass capillaries were sharpened using a flaming micropipette puller (P-1000, Sutter Instrument GmbH), filled with 10 µL of the cell suspension diluted in artificial CSF (aCSF: 126 mM NaCl, 2.5 mM KCl, 1.25 mM NaH$_2$PO$_4$, 2 mM Mg$_2$SO$_4$, 2 mM CaCl$_2$, 10 mM glucose, and 26 mM NaHCO$_3$; pH 7.4 when gassed with 95% O$_2$ and 5% CO$_2$), and fixed to the micromanipulator arm of the stereotaxic. Cell suspension was injected into the CM at a rate of 2 µL/min. At the end of the injection, mice are sutured and allowed to recover in a preheated awake cage for 1 hr, after which they are returned to the animal husbandry.

## In vivo depletion of microglia

For microglia depletion, WT and *Rag1*$^{-/-}$ mice were fed a chow diet containing the CSF1 receptor antagonist, PLX5622 (1200 ppm PLX5622 in mouse chow, Brogaarden Research Diets) for 2 weeks to induce microglia apoptosis. Mice of the control group were fed control chow diet without the antagonist.

## Infarct volume quantification

Mice were deeply anesthetized 5 days after stroke induction and transcardially perfused with 20 mL saline. Brains were removed, frozen immediately on powdered dry ice, and stored at −20°C until use. For infarct volumetry, brains were serially sectioned (400 µm intervals, 20 µm thick) and stained for cresyl violet (CV) as previously described (*Llovera et al., 2014*). CV-stained sections were scanned at 600 dpi on a fatbed scanner (Canon). Direct infarct measurement was used after validating the absence of edema at the investigated time point. The total infarct volume was measured with ImageJ and determined by integrating measured areas and distances between sections.

## Immunohistochemistry and confocal microscopy

Microglia morphology analysis was performed on brain coronal sections as previously described (*Heindl et al., 2018*). Briefly, mice were perfused with 4% paraformaldehyde (PFA), and brains were post-fixed overnight and placed in sucrose for dehydration. Then, free floating 100 µm coronal sections were stained for microglia with 1:200 anti-Iba1 (rabbit, Wako, #019–19741). Nuclei were stained using 4',6-Diamidin-2-phenylindol (DAPI, Invitrogen, #D1306), and images were acquired at a distance of 900 µm from the border of the lesion in layer 4 (ipsilateral) and the homotypic contralateral region using a Zeiss confocal microscope with 40× magnification (objective: EC Plan-Neofluar 40×/1.30 Oil DIC M27) with an image size of 1024×1024 pixel, a pixel scaling of 0.2×0.2 µm, and a depth of 8 bit. Confocal images were collected in Z-stacks with a slice-distance of 0.4 µm. Morphological features of microglia were acquired using a fully automated analysis as previously described (*Heindl et al., 2018*).

## Fluorescent in situ hybridization

Single-molecule fluorescence in situ hybridization (smFISH) was performed using the RNAscope Multiplex Fluorescent Reagent Kit v2 (Advanced Cell Diagnostics) by the manufacturer's protocols. Briefly, free floating 100 µm coronal brain sections (*Figure 3e*) or 20 µm cryo-sections (*Figure 4g*) were first dried, washed, and then incubated in RNAscope hydrogen peroxide. Antigen retrieval and protease treatment were performed as per protocol. Sections were then incubated with the probe mix (C2-*Trem2* and C1-*Cx3cr1*) for 2 hr at 40°C and then immediately washed with wash buffer. Next, sections were incubated with RNAscope Multiplex FL v2 AMP1, AMP2, and AMP3 and then probes were counterstained with TSA Plus Cy3 for C1-*Cx3cr1* and TSA Plus Cy5 for C2-*Trem2*. For microglia identification (*Figure 3e*), slides were incubated in blocking at room temperature for 1 hr before overnight incubation at 4°C with the primary rabbit anti-P2Y12 receptor antibody (1:200, AnaSpec #AS-55043A) and labeling for 1 hr with the secondary antibody AF488 goat anti-rabbit, (1:200, Invitrogen #A11034). Finally, sections were stained with DAPI (Invitrogen) and mounted with fluoromount medium (Sigma). smFISH-stained RNA molecules were counted only within the DAPI staining of the cell; a cell was considered *Cx3cr1*-positive when more than four *Cx3cr1* puncta were present.

## Whole skull immunofluorescence

Rag-1$^{-/-}$ mice were anesthetized with isoflurane and perfused transcardially with ice-cold PBS followed by 4% PFA. After removing the mandibles, skin and muscles were carefully detached from the skull (http://www.nature.com/protocolexchange/protocols/3389). The skull decalcification was performed as previously described (*Benakis et al., 2016*). Coronal skull sections (20 µm) were stained with GFP-booster Atto647N (1:500, ChromoTek GmbH) to visualize eGFP-labeled T cells. Sections were counterstained with DAPI (Invitrogen) to visualize cell nuclei and observed by confocal laser microscopy (Leica SP5).

## In vitro T cell polarization

Single-cell suspensions were generated from spleen, inguinal, axial, brachial, and mandibular lymph nodes of C57BL/6 or b-actin-EGFP mice by passing the tissue through a 70 µm cell strainer. Naive

CD4+ T cells were obtained by pre-enrichment using an 'untouched' CD4+ T Cell Isolation Kit (Miltenyi Biotec) with subsequent flow cytometric analysis (CD4+ [clone RM4-5, 0.5 ng/µL], CD44low [clone IM7, 2 ng/µL], and CD62Lhigh [MEL-14, 0.8 ng/µL]). Cells were seeded at a density of 300,000 or 400,000 cells/well in a flat-bottom 96-well plate and stimulated with plate-bound anti-CD3 and anti-CD28 Abs 0.5 µg/mL or 2 µg/mL anti-CD3 (clone 145–2 C11) for $T_{REG}$ and $T_{H1}$, respectively and 2 µg/mL anti-CD28 (clone 37.51). Different mixtures of cytokines and mAbs were added to RPMI (supplemented with 10% fetal calf serum (FCS), 50 µM β-mercaptoethanol, 50 U/mL penicillin, 50 µg/ mL streptomycin, 1% GlutaMAX, and 1% N-2-hydroxyethylpiperazine-N-2-ethane sulfonic acid [Gibco HEPES]) and used as follow: $T_{H1}$ conditions with anti-IL-4 (10 µg/mL, BioXCell, #BE0045) and IL-12 (10 ng/mL, BioLegend, #577002); $T_{REG}$ conditions: anti–IL-4 (10 µg/mL, BioXCell, #BE0045), anti-IFN-γ (10 µg/mL, BioXCell, #BE0055), and TGFβ (3 ng/mL, BioLegend, #580702). After 2 days in culture, cells were split into two new 96-well plates and incubated with freshly prepared supplemented RPMI media with IL-2 (10 ng/mL, BioLegend, #575402). Cells were cultured for a total of 5 days before injection. Quality control was performed on day 4 to assess the percentage of T cell expressing Tbet (clone 4B10, 2 ng/µL; $T_{H1}$) or FoxP3 (clone FJK-16s, 2 ng/µL; $T_{REG}$; ***Figure 3—figure supplement 1A***). One million differentiated T cells were resuspended in sterile aCSF and injected into the CM in $Rag1^{-/-}$ recipient mice 24 hr after dMCAO induction.

## IL-10 overexpression in naïve T cells

Engineered T cells overexpressing IL-10 (eTc-IL10) were generated by transfection of naïve T cells with an IL-10 plasmid (pRP[Exp]-TagBFP2-CMV>mIl10[NM_010548.2]) designed and prepared by Vector-Builder (***Figure 4—figure supplement 1B***). First, splenocytes were isolated from C57BL/6 mice (male, 6–12 weeks old) and enriched using a CD4+ T Cell Isolation Kit (Miltenyi Biotec, No:130-104-453). Quality control was performed by flow cytometry (CD4+ [clone RM4-5, 1:25], CD44low [clone IM7, 1:25], and CD62Lhigh [clone MEL-14, 1:25]). Cells were resuspended in RPMI (supplemented with 10% fetal bovine serum (FBS), 50 µM β-mercaptoethanol, 50 U/mL penicillin, 50 µg/mL streptomycin, 1% GlutaMAX, and 1% HEPES and 10 ng/mL IL-2). To induce CD4+ cells to enter the cell cycle for efficient DNA uptake, $4 \times 10^5$ cells/well were seeded in a flat-bottom 96-well plate containing bound anti-CD3 (2 µg/mL, clone 145–2 C11) and anti-CD28 (2 µg/mL, clone 37.51) for 48 hr. After 48 hr stimulation, $1.5 \times 10^6$ cells multiplied by the number of mice to be injected were transfected with the pIL-10 vector ($1 \times 10^6$ cells/1.5 µg pIL-10 DNA per cuvette) using the Mouse T Cell Nucleofector Kit (Lonza, No: VPA-1006) with Nucleofector II Device (program X-100). Once electroporated, cells were diluted with conditioned RPMI from the 48 hr stimulation and fresh supplemented RPMI (1:1) and seeded in a 12-well plate (1 cuvette of cells/ well). 24 hr post transfection, cells and supernatant were collected. Supernatant was used to confirm IL-10 secretion by ELISA (***Figure 4—figure supplement 1C***), and cells were collected for intra-CM injection ($1 \times 10^6$/mouse).

## ELISA

Secreted IL-10 was determined by ELISA as per the manufacturer's protocol (Mouse IL-10, Invitrogen, No: 88-7105-88). The color reaction was measured as OD450 units on a Bio-Rad iMark microplate reader. The concentration of supernatant IL-10 was determined using the manufacturer's standard curve over the range of 32–4000 pg/mL.

## Flow cytometry

For differentiation of live and dead cells, we stained cells with the Zombie Violet Fixable Viability Kit according to the manufacturer's instructions (BioLegend). For surface marker analysis, cell suspensions were adjusted to a density of $0.5 \times 10^6$ cells in 50 µL FACS buffer (2% FBS, 0.05% $NaN_3$ in PBS). Nonspecific binding was blocked by incubation for 10 min at 4°C with anti-CD16/CD32 antibody (Biolegend, clone 93, 5 ng/ µL) antibody and stained with the appropriate antibodies for 15 min at 4°C. The following antibodies were used for extracellular staining: CD45 (clone 30 F-11, 0.5 ng/µL), CD4 (clone RM4-5, 0.5 ng/µL), CD11b (clone M1/70, 0.6 ng/µL), CD19 (eBio1D3, 0.6 ng/µL), B220 (clone RA3-6B2, 0.32 ng/µL), CD3ε (clone 145–2 C11, 2 ng/µL), CD8a (clone 53–6.7, 2 ng/µL), and CD62L (clone MEL-14, 0.8 ng/µL) from Thermofisher. For intracellular cytokine staining, cells were restimulated for 4 hr with PMA (50 ng/mL, Sigma), ionomycin (1 µM, Sigma), and brefeldin A (1 µL for ~$10^6$ cells/mL). Cells were then stained for surface markers as detailed below, fixed, and permeabilized using Fixation

and Permeabilization Buffers from eBiosciences following the manufacturer's instructions. Briefly, cells were fixed for 30 min at 4°C (or RT for FoxP3), washed with permeabilization buffer, and incubated for 30 min with the appropriate antibodies in permeabilization buffer at 4°C (or RT for FoxP3). The cells were stained with the transcription factors FoxP3 (clone FJK-16s, 2 ng/µL) and T-bet (clone 4B10, 2 ng/µL) or IFN-γ (clone 4 S.B3, 2 ng/µL). Cells were washed with FACS buffer, resuspended in 200 µL of FACS buffer and acquired using a BD FACSverse flow cytometer (BD Biosciences, Germany), and analyzed using FlowJo software (Treestar, USA). Isotype controls were used to establish compensation and gating parameters.

## Nanostring analysis

The ipsilateral hemispheres were lysed in Qiazol Lysis Reagent, and total RNA was extracted using the MaXtract High Density kit with further purification using the RNeasy Mini Kit (all Qiagen). 70 ng of total RNA per sample was then hybridized with reporter and capture probes for nCounter Gene Expression code sets (Mouse Neuroinflammation codeset) according to the manufacturer's instructions (NanoString Technologies). Samples (6/condition) were injected into NanoString cartridge, and measurement run was performed according to nCounter SPRINT protocol. Background (negative control) was quantified by code set intrinsic molecular color-coded barcodes lacking the RNA linkage. As a positive control code set, intrinsic control RNAs were used at increasing concentrations. Genes below the maximal values of the negative controls were excluded from the analysis. All gene counts were normalized (by median) and scaled (mean-centered and divided by SD of each variable). Heatmaps were performed using the MetaboAnalystR package on normalized expression values. The regulated genes in microglia treated with $T_{H1}$ or $T_{REG}$ in comparison to vehicle treated microglia (CT) are represented in the volcano plots; genes with a p<0.05 were color-coded. The significantly up-regulated genes in microglia (FC>1.5 and p<0.05) were further used for pathway analysis using Cytoscape ClueGO (*Bindea et al., 2009*): $T_{H1}$/CT, 19 up-regulated genes (*Axl, Cd74, Cryba4, Cxcl9, Ezh1, Fgl2, Gbp2, Klrk1, Irf7, Klrk1, Lag3, Map1lc3a, Nkg7, Pld2, Setd7, Siglec1, Stat1, Tnfsf10, Ttr,* and *Zbp1*) and $T_{REG}$/CT, 8 up-regulated genes (*Ccl2, Ccl7, Cd69, Clcf1, Cxcl10, Irf7, Nkg7,* and *Siglec1*).

## Microglia cell isolation for RNA sequencing

Mice were perfused transcardially with ice-cold saline containing Heparin (2 U/mL). Brains were placed in HBSS (w/ divalent cations $Ca^{2+}$ and $Mg^{2+}$) supplemented with actinomycin D (1:1000, 1 mg/mL, Sigma, #A1410), and microglia was isolated with the Papain-based Neural Tissue Dissociation Kit (P) (# 130-092-628, Miltenyi Biotec B.V. & Co. KG) according to the manufacturer's instructions. Cell suspension was enriched using 30% isotonic Percoll gradient. $1\times10^3$–$1.5\times10^3$ live microglia cells from 3 mice per condition were sorted according to their surface marker CD45+CD11b+7-AAD negative (SH800S Cell Sorter, Sony Biotechnology) and proceed for 10× Genomics according to the manufacturer's instructions (ChromiumTM Single Cell 3' Reagent kits v2).

## Single-cell data analysis

The CellRanger software (v2.0.0, 10× Genomics) was used for demultiplexing of binary base call files, read alignment, and filtering and counting of barcodes and unique molecular identifiers (UMIs). Reads were mapped to the mouse genome assembly reference from Ensembl (mm10/GRCm38). Downstream data analyses were performed using the Scanpy API (scanpy v ≥ 1.4 with python3 v ≥ 3.5; *Wolf et al., 2018*). Details on analyses, selected thresholds, and package versions are provided in available source scripts (See Code and Data availability). Outlier and low-quality cells were filtered if the fraction of mitochondria-encoded counts was greater than 10%, or the total number of counts was greater than 48,000. Thresholds were selected upon visual inspection of distributions as recommended (*Luecken and Theis, 2019*). Genes expressed in less than 10 cells were excluded. Furthermore, doublet cells as identified by the Scrublet algorithm (v0.2.1; *Wolock et al., 2019*) were excluded. Doublet scores and thresholds were determined for each sample separately. Raw counts of a cell were normalized by total counts neglecting highly expressed genes which constitute more than 5% of total counts in that cell. Then, counts were log-transformed (log[count+1]). These processed and normalized count matrices were used as input for all further analyses.

For the full data set and the microglia subset, first a single-cell nearest-neighbor graph was computed on the first 50 independent principal components. Principle components were calculated

using the 3000 most variable genes of the full data set as input. The UMAP algorithm (*Becht et al., 2019*) as used to obtain a two-dimensional embedding for visualization. Iterative clustering was performed with the Louvain algorithm (*Blondel et al., 2008*) as implemented in louvain-igraph (v0.6.1, Traag et al., https://github.com/vtraag/louvain-igraph) with a varying resolution parameter. Clusters were annotated using previously described marker genes and merged if expressing the same set of marker genes. For *Figure 2d and e*, data were converted into a Seurat object and further analyzed in R to identify differentially expressed genes between WT and *Rag1*⁻/⁻ samples in stroke and naïve conditions (Seurat package, version 4.2.0 *Hao et al., 2021*). Gene dataset associated with microglia subsets was submitted to log-normalization, identification of high-variable genes using the mean-variance plot (MVP) method, scaling, and regression against the number of UMIs and mitochondrial RNA content per cell. Data was further subjected to unsupervised clustering and embedded using UMAP. Differentially expressed genes between WT and *Rag1*⁻/⁻ samples in stroke and naïve conditions were calculated using the FindMarkers function. Volcano plots were created using EnhancedVolcano in R (*Blighe et al., 2022*). To obtain the Venn diagram, the significantly regulated genes (adjusted p-value<0.05) with a fold change of 1.5 (Log2FC = 0.6) – excluding Gm genes, mitochondrial and ribosomal genes (*Mrpl*, *Mrps*, *Rpl*, and *Rps*) – were included in the analysis. The differentially expressed genes that were regulated after stroke in comparison to naïve condition WT or *Rag1*⁻/⁻ included 84 genes and 187 genes, respectively.

Trajectories from homeostatic to reactive microglia were inferred with PAGA (*Wolf et al., 2018*) and diffusion pseudotime (DPT; *Haghverdi et al., 2016*) algorithms. First, clusters were grouped into two paths connecting the root and end cell cluster based on the computed cluster connectivities (PAGA), then cells were ordered along these paths based on the random-walk-based cell-to-cell distance (DPT). To capture processes specific to the path 2 trajectory in stroke-associated microglia, data was first subset to cells of path 2 and end cells clusters of stroke samples and gene expressed in less than 20 cells of the subset excluded. Then, gene sets were computed by clustering the 500 most varying genes using their pairwise-Pearson correlation values as input and Ward's hierarchical clustering method with Euclidean distance (scipy python package v.1.5.4; *Virtanen et al., 2020*). One gene set with average correlation <0.05 was excluded. Finally, to obtain an activation score per cell for a given gene set, cell scores were computed as described by *Satija et al., 2015* and implemented in Scanpy in the *tl.score_genes* functionality. Differential activation of gene sets between WT and *Rag1*⁻/⁻ samples was determined by a Wilcoxon rank sum test. To identify genes differentially regulated along the inferred cellular trajectory, a differential gene expression test (Welch t-test with overestimated variance) between the root and end cell cluster was performed for WT and *Rag1*⁻/⁻ samples separately. Non-overlapping, significantly changing genes (p-value<0.05 corrected for multiple testing with the Benjamin-Hochberg method) were considered as regulated specifically in WT and *Rag1*⁻/⁻ samples, respectively. Pathway enrichment of gene sets and differentially regulated genes was performed with the gseapy package (https://github.com/zqfang/GSEApy/) functionality of EnrichR (*Xie et al., 2021*).

## Code availability

Jupyter notebooks with custom python scripts for scRNA-seq analysis is available in a github repository (https://github.com/Lieszlab/Benakis-et-al.-2022-eLife.git), copy archived at swh:1:rev:04f5dead-312f071a4c760607d68f94047444bbaa (*Benakis et al., 2022*).

## Statistical analysis

Data are expressed as mean ± SD or median with interquartile range and were analyzed by unpaired Student's t-test (two-tailed) or one- or two-way ANOVA and post-hoc tests as indicated in the figure legends. Exclusion criteria are described in the individual method sections. The data for microglia morphology are shown as median ± interquartile range, and statistical significance was tested using the Wilcoxon rank sum test with continuity correction and Bonferroni post-hoc correction for multiple testing in R (version 4.0.3).

## Acknowledgements

The authors thank Kerstin Thuβ-Silczak and Dr. Monica Weiler for technical support and Michael Heide and Oliver Weigert for support at the DKTK Nanostring core facility. Some of the graphical schemes

were created with BioRender.com. The study was supported by the European Research Council (ERC-StG 802305) and the German Research Foundation (DFG) under Germany's Excellence Strategy (EXC 2145 SyNergy – ID 390857198), through SFB TRR 274, FOR2879 and under the DFG projects 405358801 (to A.L.), 418128679 (to C.B.) and PE-2681/1–1 (to A.P.).

# Additional information

## Competing interests

Fabian J Theis: reports receiving consulting fees from ImmunAI and ownership interest in Dermagnostix. The other authors declare that no competing interests exist.

## Funding

| Funder | Grant reference number | Author |
|---|---|---|
| European Research Council | ERC-StG 802305 | Arthur Liesz |
| Deutsche Forschungsgemeinschaft | EXC 2145 SyNergy - ID 390857198 | Arthur Liesz |
| Deutsche Forschungsgemeinschaft | SFB TRR 274 | Arthur Liesz |
| Deutsche Forschungsgemeinschaft | FOR2879 | Arthur Liesz |
| Deutsche Forschungsgemeinschaft | 405358801 | Arthur Liesz |
| Deutsche Forschungsgemeinschaft | 418128679 | Corinne Benakis |
| Deutsche Forschungsgemeinschaft | PE-2681/1-1 | Anneli Peters |
| Deutsche Forschungsgemeinschaft | CRC TRR 355 (ID 490846870) | Arthur Liesz Stefan Bittner Corinne Benakis |

The funders had no role in study design, data collection and interpretation, or the decision to submit the work for publication.

## Author contributions

Corinne Benakis, Conceptualization, Formal analysis, Supervision, Funding acquisition, Investigation, Visualization, Methodology, Writing - original draft, Writing - review and editing; Alba Simats, Supervision, Validation, Investigation, Methodology; Sophie Tritschler, Simon Besson-Girard, Data curation, Formal analysis, Visualization; Steffanie Heindl, Data curation, Formal analysis, Investigation; Gemma Llovera, Kelsey Pinkham, Formal analysis, Investigation, Methodology; Anna Kolz, Alessio Ricci, Investigation, Methodology; Fabian J Theis, Data curation, Supervision; Stefan Bittner, Writing - review and editing; Özgün Gökce, Data curation, Formal analysis, Writing - review and editing; Anneli Peters, Investigation, Methodology, Writing - review and editing; Arthur Liesz, Conceptualization, Supervision, Funding acquisition, Writing - original draft, Project administration, Writing - review and editing

## Author ORCIDs

Corinne Benakis ![ORCID] http://orcid.org/0000-0001-6463-7949
Steffanie Heindl ![ORCID] http://orcid.org/0000-0003-3576-2702
Simon Besson-Girard ![ORCID] http://orcid.org/0000-0003-1194-5256
Anna Kolz ![ORCID] http://orcid.org/0000-0002-6020-7746
Alessio Ricci ![ORCID] http://orcid.org/0000-0002-3051-8113
Arthur Liesz ![ORCID] http://orcid.org/0000-0002-9069-2594

## Ethics

All animal procedures were performed in accordance with the guidelines for the use of experimental animals and were approved by the respective governmental committees (Licenses: 02-21-46 and 02-21-95; Regierungspraesidium Oberbayern, the Rhineland Palatinate Landesuntersuchungsamt Koblenz). All animal experiments were performed and reported in accordance with the ARRIVE guidelines (Kilkenny et al., 2011).

## Decision letter and Author response

Decision letter https://doi.org/10.7554/eLife.82031.sa1
Author response https://doi.org/10.7554/eLife.82031.sa2

## Additional files

### Supplementary files

• MDAR checklist

### Data availability

Data is available in github repository (https://github.com/Lieszlab/Benakis-et-al.-2022-eLife, copy archived at swh:1:rev:04f5dead312f071a4c760607d68f94047444bbaa).

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
