## [Editor Report]

This manuscript should be of interest to neuroimmunologists investigating how microglia may be manipulated to improve neuroinflammation in stroke and beyond. The data support the hypothesis that manipulation of lymphocytes and the cytokines they secrete may be an effective therapeutic strategy to modulate inflammation and improve the outcome after stroke.

---

## [Decision Letter]

**Decision letter after peer review:**

[Editors’ note: the authors submitted for reconsideration following the decision after peer review. What follows is the decision letter after the first round of review.]

Thank you for submitting the paper "T cells modulate the microglial response to brain ischemia" for consideration by *eLife*. Your article has been reviewed by 3 peer reviewers, one of whom is a member of our Board of Reviewing Editors, and the evaluation has been overseen by a Reviewing Editor and a Senior Editor. The following individual involved in review of your submission has agreed to reveal their identity: Jane Foster (Reviewer #1).

Comments to the Authors:

Although there are several interesting findings, more in depth analyses is needed and underlying mechanism and functional link between microglia and stroke pathology are needed. A more comprehensive histological assessment (possibly a time course) of microglia and invading T cells that showed regional and cellular localization would strengthen the evidence of T cells involvement and impact on microglia.

Summary of key comments (see detailed reviewer comments below)

1. The authors cannot conclude that the differences in microglia seen in RAG2-/- are solely attributable to T cells from the data provided in the manuscript. Pease include gene expression data from naive WT and RAG2-/- animals and could include this to demonstrate any baseline effects of the model on microglia.

2. Re Figure 1 and Figure 2: The comparison of gene expression changes between WT and Rag2KO microglia is difficult to interpret because they are displayed as changes in gene modules rather than changes in expression of individual genes. Are there individual genes that change significantly in WT vs KO microglia after stroke, and if so data need to be validated by an orthogonal method (e.g. flow cytometry, qPCR, etc).

How do these changes relate to a potential role for microglia in the pathology of stroke? These considerations and validations will be necessary to make sense of the sequencing data in the context of this manuscript.

3. Re: Figure 2 and Figure 3: It is unclear what cells the authors have sorted into pools and profiled using SmartSeq 2 (Figure 2) or for the nanostring analyses after T cell injection (Figure 3). If their sorting strategy was the same as for the 10x experiments (CD45+CD11b+ cells), then they are likely looking at a mix of microglia and other myeloid cells in each pool (see comments below).

4. Related to Figure 4: The result that adoptive transfer of IL-10-expressing T cells reduces forepaw asymmetry after stroke is promising and warrants further investigation into the potential mechanisms. The potential link to microglia could be tested by depletion of microglia (e.g. by PLX5622), or more elegantly by adoptively transferring IL-10 producing T cells into mice where the IL-10 receptor is ablated in microglia/myeloid cells (e.g. Cx3cr1-CreERT2;Il10ra-flox mice).

5. Given the differential involvement of lymphocyte subsets in stroke pathophysiology it is very important to know any impact the lack of lymphocytes may have on factors such as stroke lesion size and infiltration of other immune cells which could skew the microglial profile.

Essential experiments:

1. Comparison of gene expression between WT and RAG2-/- in naive animals.

2. Pathology analysis of stroke lesions in experimental models

3. Flow cytometry to determine the numbers of neutrophils, monocyte and macrophages present in the brain in experimental models.

4. Care must be taken in text to identify where non-microglial cells may be contributing to effects observed.

*Reviewer #1 (Recommendations for the authors):*

The authors explored the role of T cells in the microglial response to experimental stroke. Using several genetic, cellular, and molecular approaches, they provide evidence that the microglial response can be influenced by T cell signaling. The authors suggest that T cell-related treatment may have a role in preventing the secondary damage that occurs following stroke.

Strengths:

1. The study employs single-cell sequencing to specifically examine microglial and invading macrophage gene expression profiles following experimental stroke.

2. The importance of specific T cell populations was tested using adoptive transfer of Th1 and Treg cells. This approach demonstrates that Th1 T cells can result in upregulation of microglial activation genes, and that Treg cells can result in a more protective gene expression activation pattern.

3. The authors used a combination of techniques to identify and verify the gene expression patterns linked to T cell-microglial interactions.

Weaknesses:

1. Rag-1 deficient mice lack mature T cells and B cells. They are an excellent model to study the role of the immune system in disease models but findings are not specific to T cells.

2. The microglial morphological differences shown in Figure 1C are significant but not distinct profiles. In addition, the profile of reactive and homeostatic microglia are not distinguished between WT and Rag-1 deficient mice. Several of the follow up observations are noted as significant but this is not visually evident.

3. The adoptive transfer experiments were conducted only in RAG^-/-^ mice so there is not supporting evidence that this T cell-related response would occur in a WT context.

4. The rationale for the IL-10 engineered T cells is not provided. The experimental flow does not extend to this experiment, and yet, it is the primary topic of the discussion.

5. The reliance on bioinformatic pathway analysis without further direct validation limits the impact of the findings.

6. The conclusion that specific T cells drive the polarization of microglia in response to experimental stroke is suggested from the results but not demonstrated.

A clear overview of the timeline and schematic of the different experiments is needed. It is not clear that the authors integrated the findings across the different experimental approaches. Many genes are included in the narrative but a consensus of the cellular and molecular mechanisms that are participating is not clear.*Reviewer #2 (Recommendations for the authors):*

In this study the authors used transcriptomic analysis of microglia in a lymphocyte deficient mouse (Rag1-/-) to understand the role of T cell-microglial interactions in neuroinflammation induced by experimental stroke. The authors determine microglia were less activated by morphology and showed altered gene expression after stroke, when lymphocytes were not present. T cells that were polarised to TH1 or Tregs, which have differential cytokine secretion profiles, were then injected into these animals after experimental stroke to determine the effect on microglia. This demonstrated the differential effects of these T cell subsets and their associated cytokines on microglia after stroke. The therapeutic potential of T cells that over express IL-10 was then investigated and the authors found no difference in the extent of the brain injury but an improvement in motor function in treated animals.

Datasets from the single cell RNAseq and smart-seq2 experiments will be publicly available and will be a useful resource for researchers interested on the effects of stroke on microglia in the presence or absence of lymphocytes. The Rag1-/- model used has no mature B or T lymphocytes from birth. The authors show that in the contralateral hemisphere of the brain there are no morphological differences in microglia between Rag1-/- and WT animals. Given our recent understanding of lymphocytes in the meninges and how they contribute to development of some brain cell populations and play a role in both CNS homeostasis and disease, it would be useful to see an analysis of gene expression changes between WT and Rag1-/- in naive animals to understand if there are any baseline effects on microglia in this model. Figure 2f indicates that there are baseline effects in gene cluster 6 in naive animals and this cluster has moderate expression across all microglial samples identified (Figure 2e) with the highest expression in sample 2 which is highly regulated by stroke (Figure 2d).

Microglia from lymphocyte deficient mice showed increased cytokine production and altered chemokine expression. Lymphocytes, in particular T cells, are thought to contribute to the extent of the brain injury after stroke. It would be useful to know if the size of the lesions produced in Rag1-/- animals was reduced in comparison to those in WT to determine if reduced injury (and therefore less DAMPs) also could contribute to the altered microglia phenotype.

Neutrophil degranulation was identified as the most highly upregulated pathway in microglia in Rag1-/- mice after stroke. As chemokine expression was also altered, the number and composition of immune cells infiltrating into the brain after stroke may be different. Alterations to the number or proportion of neutrophils and monocytes/ macrophages in the brain after stroke could also impact microglial gene expression. It would be useful to know the post-stroke immune environment in the RAG1-/- mice to fully interpret microglial gene expression changes.

Purified TH1 or Treg were then injected into the cisterna magna 24 h after stroke and nanostring analysis was used to examine microglial expression of genes associated with neuroinflammation. This experiment nicely showed that purified T cell populations in a lymphocyte deficient background are able to influence microglial gene expression and morphology. Again the impact of these treatments on the extent of brain injury should also be considered.

The authors then investigated the therapeutic potential of Treg which overexpress IL-10 in experimental stroke. The T cells were injected at a translationally relevant time, 4 h post experimental stroke, and the experiment was carried out in WT animals in the context of the total neuroinflammatory response to stroke. Nanostring was again used to determine neuroinflammatory gene expression however in this instance it was performed on RNA from the whole brain hemisphere and not sorted microglia. Many of the differentially expressed genes will also be expressed in infiltrating monocytes/ macrophages and therefore cannot be solely attributed to altered microglial phenotype. However the positive impact on motor function in the animals that received these cells is encouraging.

The authors conclude that T cells are capable of modulating microglial polarisation in the context of stroke mainly via the secretion of cytokines. It should be noted that T cells are not the only immune cell type capable of producing these cytokines after stroke. In particular, regulatory B cell derived IL-10 is known to reduce brain injury after stroke and may act on microglia in a similar manner. The use of engineered T cells to generate a stable, local production of IL-10 in the injury site shows promise. In future development of this work it would be interesting to see the effect on further behavioral and functional assessments and the spread of microglial activation to sites anatomically distinct, but functionally connected, to the primary infarct.

The following data would strengthen the manuscript and allow a full interpretation of which factors may be influencing microglial phenotype in this manuscript.

1. Analysis of differentially expressed genes in naive WT and Rag1-/- microglia to understand baseline effects of the absence of lymphocytes in microglia.

2. Quantification of infarct sizes in Rag1-/- and WT animals after stroke.

3. Flow cytometry analysis of immune cell populations present in the brain at 5d post stroke to determine the number and proportion of neutrophils, monocytes and macrophages in RAG^-/-^ and WT animals.

4. Image analysis treating an individual microglia as a data point is pseudoreplication. This should be averaged within one animal, which is the experimental unit in this design, and analysed appropriately.

5. The n number of animals going into experiments is not always clear in figure legends.

In the discussion, it should be acknowledged that T cells are not the only immune cell that may influence microglial polarisation in this manner. IL-10 producing regulatory B cells should be discussed.

The authors should justify why no sham-operated controls were used in these experiments.

The authors should discuss the use of RNA extracted from whole brain hemisphere in the final experiment and highlight that cells that are not microglia may contribute to gene expression changes identified.

*Reviewer #3 (Recommendations for the authors):*

In this study, the authors investigate the contributions of microglia -T cell interaction in in modulating stroke pathology using several approaches. Microglia were profiled from WT and Rag2-KO mice after stroke and using adoptive transfer of differently polarized (Th1 or Treg) T cells or T cells engineered to express IL-10 into the cisterna magna of Rag2-KO mice after stroke to assess alterations in microglia and stroke pathology. Although there are several interesting findings, more in depth analyses is needed and underlying mechanism and functional link between microglia and stroke pathology are unclear.

The authors claim that microglia from Rag2KO mice respond differently to stroke than WT microglia, though it is difficult to ascertain what these changes are, the magnitude of these changes, and how they might be relevant to the role of microglia in stroke from the data presented. In some experiments, the sequencing data also suffers from a lack of specificity (e.g. it appears that the authors are studying a broad pool of mixed myeloid cells and calling them "microglia"), which substantially weakens any of the conclusions the authors draw about microglia in the paper. Further, the authors do not test the potential functional link between microglia and stroke pathology (i.e. by depletion of microglia). In the absence of this data, it is unclear whether T cell modulation of microglia even has the potential to influence stroke outcomes. The data indicating that adoptive transfer of IL-10-expressing T cells improves behavioral outcomes after stroke is intriguing. The authors should consider expanding on this point and studying the mechanisms(s) by which this might work. This could be dependent on or independent of microglia; and further analysis of this point would substantially strengthen this manuscript.

1) Related to Figure 1 and Figure 2: The comparison of gene expression changes between WT and Rag2KO microglia is difficult to interpret because they are displayed as changes in gene modules rather than changes in expression of individual genes. Are there individual genes that change significantly in WT vs KO microglia after stroke, and if so can these changes be validated by an orthogonal method (e.g. flow cytometry, qPCR, etc)? How do these changes relate to a potential role for microglia in the pathology of stroke? These considerations and validations will be necessary to make sense of the sequencing data in the context of this manuscript.

2) Related to Figure 2 and Figure 3: It is unclear what cells the authors have sorted into pools and profiled using SmartSeq 2 (Figure 2) or for the nanostring analyses after T cell injection (Figure 3). If their sorting strategy was the same as for the 10x experiments (CD45+CD11b+ cells), then they are likely looking at a mix of microglia and other myeloid cells in each pool. This would make sense given some of the pathways that appear to be altered (e.g. neutrophil degranulation in the SmartSeq2 data). While this may be interesting, it is incorrect to attribute these changes in gene expression to changes in microglia as the authors do throughout both figures. This interpretation should be corrected, or the experiments repeated with a more specific sorting strategy if the authors wish to make any claims about alterations in microglia in either of these experiments (e.g. could sort microglia as CD45+, CD11b+, GR-1-, CD64+, P2RY12 or CX3CR1high, CD206low).

3) Related to Figure 4: The result that adoptive transfer of IL-10-expressing T cells reduces forepaw asymmetry after stroke is promising and warrants further investigation into the potential mechanisms. The potential link to microglia could be tested by depletion of microglia (e.g. by PLX5622), or more elegantly by adoptively transferring IL-10 producing T cells into mice where the IL-10 receptor is ablated in microglia/myeloid cells (e.g. Cx3cr1-CreERT2;Il10ra-flox mice).

Additionally, is there a potential impact on other cell types that have been shown to be important in recovery from stroke? Some discussion of this point would be helpful.

1. Related to Figures 1c, 3e-f, 4g: It is not appropriate to use individual microglia as n values when quantifying microglia morphology, RNAscope puncta in microglia, etc. Instead, mice should be used as biologically independent replicates. It does not appear that Trem2 expression differences in Figure 3e and 4g would be anywhere near statistically significant given the variation if appropriate statistics were applied. Similarly, we have concerns about the statistics applied to the SmartSeq2 data presented in Figure 2f. The authors should use a differential expression method designed specifically for single cell sequencing data to compare WT to KO microglia. Again, using individual cell pools as n values in this case when running statistical tests massively inflates the resulting p value and makes very small changes in gene expression erroneously appear significant.

[Editors’ note: further revisions were suggested prior to acceptance, as described below.]

Thank you for resubmitting your work entitled "T cells modulate the microglial response to brain ischemia" for further consideration by *eLife*. Your revised article has been evaluated by Carla Rothlin (Senior Editor) and a Reviewing Editor.

This manuscript should be of interest to neuroimmunologists investigating how microglia may be manipulated to improve neuroinflammation in stroke and beyond. The data support the hypothesis that manipulation of lymphocytes and the cytokines they secrete may be an effective therapeutic strategy to modulate inflammation and improve outcome after stroke.

The manuscript has been improved but there are some remaining issues that need to be addressed. As you revise your manuscript, please take note of the comments below, in particular, those from Rev. 1

*Reviewer #1 (Recommendations for the authors):*

The authors have attempted to address previous comments and have done so in some cases. There are still several important controls missing that would allow them to make the link between T cells and microglia phenotype/function in stroke.

Addition of functional evidence to support this link is needed to support the claims – otherwise showing how T cells polarize microglia transcriptionally after stroke doesn't seem like a substantial advance in the field. Also, the fact that deleting microglia makes the infarct worse is confusing and makes it really difficult to understand how T cells program microglia to cause worse pathology later. It is likely that microglia play both protective and damaging roles, but they would need more elegant experiments or manipulation of specific pathways in microglia to tease that apart.

Specific Comments:

1. The authors now show changes between WT and Rag2KO myeloid cells after stroke as individual genes in addition to pathways. This addresses our comment. However, since this is single cell sequencing data, the authors should make DE gene expression comparisons within individual cell types (e.g. WT vs KO microglia) instead of pooling data across cell types.

2. We appreciate the addition of the microglia depletion experiment in figure 1g. However, this experiment still does not address the question of whether T cell-microglia interaction affects stroke pathology because it is missing the WT control and depletion conditions. From the data presented, all we can conclude is that microglia influence stroke pathology on a Rag2KO background. Depletion of microglia may well have the same effect in WT mice, which would argue against a role for T cell priming of microglia function.

3. In general, the authors need to revise their discussion of the role of T cells in influencing microglia "function" after stroke. Without clear experiments linking T cells -> microglia -> pathology, or alternatively T cells to modulation of a functional readout in microglia (e.g. engulfment), the authors can only conclude that T cells modulate microglia morphology and transcriptome as they have shown throughout the manuscript.

e.g. line 340: "Here, we established a mechanistic link between T cells and microglial function and showed the distinct role of T cell subpopulations on switching microglial polarization state in response to stroke."

*Reviewer #2 (Recommendations for the authors):*

The authors have addressed the comments and conducted additional experiments that have improved the manuscript. The edited results narrative provides a better guide to the key findings and the additional results allow better interpretation of the original findings.

It would be good to update the Summary and Abstract to highlight key take homes. In particular, the impact of the stroke itself on lesion size, as well as the particular potential protective or detrimental role for specific T cells.

*Reviewer #3 (Recommendations for the authors):*

The authors have addressed the majority of concerns with the manuscript and the new analyses included allow for a more transparent interpretation of the data within.

Gene expression changes between naive WT and Rag1-/- mice, flow cytometry of innate immune cells in WT and Rag1-/- after stroke and information on stroke lesion size all help to exclude non-T cell related factors that may influence microglial gene expression. The linear mixed model analysis, with grouping as a variable, improves the statistical analysis performed on microglial morphology data.

Furthermore, the alterations to the text have reduced over interpretation of the data in the manuscript and better reflect the data shown.

Although the authors were unable to perform an experiment using IL-10R -/- microglia due to difficulties in receiving the appropriate mouse lines, the final experiment in the manuscript shows that transfer of IL-10 over expressing T cells had a biological effect on microglia in the context of a conventional neuroinflammatory response to stroke in WT mice. Therefore although direct signalling between the IL-10+ T cells and the microglia has not been definitively proven, the resulting effects on microglial gene expression and stroke functional outcome show this to be a promising therapeutic strategy.

I believe the authors have satisfied the reviewers original comments with no further changes required.

---

## [Author Response]

[Editors’ note: the authors resubmitted a revised version of the paper for consideration. What follows is the authors’ response to the first round of review.]

Comments to the Authors:Although there are several interesting findings, more in depth analyses is needed and underlying mechanism and functional link between microglia and stroke pathology are needed. A more comprehensive histological assessment (possibly a time course) of microglia and invading T cells that showed regional and cellular localization would strengthen the evidence of T cells involvement and impact on microglia.Summary of key comments (see detailed reviewer comments below)1. The authors cannot conclude that the differences in microglia seen in RAG2-/- are solely attributable to T cells from the data provided in the manuscript. Pease include gene expression data from naive WT and RAG2-/- animals and could include this to demonstrate any baseline effects of the model on microglia.

As suggested, we added the analysis of the differential gene expression from naive WT and Rag1^-/-^animals (new Figure 2b). Additionally, we included changes in the text addressing that in the Rag-1 deficient mouse model microglia changes are not solely attributed to mature T cells, but also could be due to B cell deficiency (see point 4 below). Furthermore, we performed flow cytometry analysis of the brain myeloid cells in WT and Rag1^-/-^ (new Suppl. Figure 3b and detailed response in point 4).

Based on these additional analyses and based on published results, we believe that in stroke the effect of the lymphocyte-deficiency in Rag1^-/-^ mice is mainly due to T cells. Indeed, published findings showed that by comparing Rag1^−/−^ mice reconstituted with B cells to Rag1^−/−^ mice developed significantly smaller brain infarctions similar to Rag1^-/-^, whereas Rag1^−/−^ mice reconstituted with CD3+ T cells had similar large infarcts as WT mice (Kleinschnitz et al., 2010), showing that B cells have little impact on the early development on the brain infarct. In addition, we generated new data for the revision of our manuscript to support this point (Suppl Figure 3). We performed flow cytometry analysis of infiltrating leukocytes isolated from the ipsilesional hemisphere 5 days after dMCAO in wild-type (WT) and Rag-1 deficient mice. First, these new data demonstrate that in WT mice, T cells are 14 times more abundant than B cells in the ipsilesional hemisphere (Suppl. Figure 3a), also supported by others (Gelderblom et al., 2009). In addition, the abundance of myeloid cell subsets is not affected by the Rag1 gene deletion (Suppl. Figure 3b). Together, these data support the hypothesis that T cells are the main contributor to the observed microglial phenotype.

2. Re Figure 1 and Figure 2: The comparison of gene expression changes between WT and Rag2KO microglia is difficult to interpret because they are displayed as changes in gene modules rather than changes in expression of individual genes. Are there individual genes that change significantly in WT vs KO microglia after stroke, and if so data need to be validated by an orthogonal method (e.g. flow cytometry, qPCR, etc).How do these changes relate to a potential role for microglia in the pathology of stroke? These considerations and validations will be necessary to make sense of the sequencing data in the context of this manuscript.

Microglia function in the context of stimulus-evoked activation and in response to tissue injury represents a complex adaption to the tissue micromilieu and can not be solely attributed to a single gene (or receptor, or pathway). Correspondingly, a large set of altered genes in response to stroke indicates the substantially altered cellular function of microglia in response to stroke. Therefore, the use of transcriptomic analyses in this context is rather to describe functional gene sets as an indicator of the cellular state rather than a tool to study single gene modifications or specific pathways. This approach to using transcriptomic analyses of microglia has been essential for studying the functional heterogeneity in numerous recent studies explaining distinct microglial functions in pathology – such as “disease-associated MG” (DAMs), “white-matter associated MG” (WAMs), and others – by describing gene sets instead of single regulated genes (Hammond et al., 2019; Keren-Shaul et al., 2017; Safaiyan et al., 2021). In our study, we aimed to test as a proof-of-concept the impact of T cells on altering the microglial response to stroke. In this regard the regulation of single genes in microglia is not the primary goal of this research, because many gene functions are redundantly regulated in microglia, the transcriptomic landscape in general is altered upon stroke and rather gene sets (representing functional pathways) than single genes are regulating gene function. Nevertheless, we report now results on individual differentially regulated genes between naïve WT and Rag^−/−^ and stroke WT and Rag^−/−^ (new Figure 2b).

Besides this approach to use larger gene sets to describe a microglial subset or function, our analysis also identified several differentially regulated genes which have been well characterized to contribute to stroke pathology. For example, the regulated chemokines CCL2, CCL7 and CXCL10 are well known to be key chemoattractant signals for the recruitment of circulating immune cells to the ischemic brain but can also represent guiding cues for neuronal stem cells important in the post-stroke recovery phase (Andres et al., 2011; Dimitrijevic et al., 2007; Li et al., 2020). Similarly, we observed regulation of various molecules which are associated with or part of the major histocompatibility process (H2-ab1, CD74, and other H2 molecules). MHC-dependent processing and presentation of (self-)antigens after stroke is a key function of microglia as antigen-presenting cells in the CNS and might contribute to the reciprocal interaction with lymphocytes and their activation (Berchtold et al., 2020).

The “validation” of single-cell sequencing data has been a continuous issue of discussion in the past years. In our perception the validation of single-cell sequencing data by FACS or PCR as suggested cannot confirm the scSeq findings in a robust way. Unfortunately, antibodies for quantitative assessment of protein expression levels per cell are not available for most regulated genes. PCR is an inferior method in terms of sensitivity, specificity, and detection robustness in comparison to single-cell sequencing using the 10x scSeq pipeline. An overview of this ongoing problematic and why orthogonal “validation” tools are not appropriate anymore in the era of highly standardized and sensitive sequencing pipelines is given for example in a recent editorial by Tom Coenye (Coenye, 2021). For these reasons, we believe it is more informative and robust to present transcriptional changes as gene sets than single gene to be validated by less power methods.

3. Re: Figure 2 and Figure 3: It is unclear what cells the authors have sorted into pools and profiled using SmartSeq 2 (Figure 2) or for the nanostring analyses after T cell injection (Figure 3). If their sorting strategy was the same as for the 10x experiments (CD45+CD11b+ cells), then they are likely looking at a mix of microglia and other myeloid cells in each pool (see comments below).

In the revised manuscript, more detail is given now how the cells were sorted for the Nanostring analysis (related to Figure 3). We also clarified in the revised text, that the results might be confounded by other cell populations than microglia:

“Although the observed regulated genes are well known to be associated with microglial function, it is conceivable that in this analysis other cell types than microglia, including various brain-invading myeloid cell subsets, could account for this effect since the whole ischemic brain tissue was processed for Nanostring analysis.” (Lines 302-305, pages 11-12).

4. Related to Figure 4: The result that adoptive transfer of IL-10-expressing T cells reduces forepaw asymmetry after stroke is promising and warrants further investigation into the potential mechanisms. The potential link to microglia could be tested by depletion of microglia (e.g. by PLX5622), or more elegantly by adoptively transferring IL-10 producing T cells into mice where the IL-10 receptor is ablated in microglia/myeloid cells (e.g. Cx3cr1-CreERT2;Il10ra-flox mice).

As suggested, we performed a new experiment aiming at addressing the lack of microglia and its implication in microglia–T cell interactions on the development of the infarct lesion. Since the lack of lymphocytes induces morphological changes of microglia towards a less activated state, we hypothesized that the lack of microglia would be neuroprotective. We depleted microglia using PLX5622 in Rag1^-/-^ mice. Surprisingly, we found a significant increase of the infarct volume in Rag1^-/-^ mice depleted of microglia in comparison to vehicle-treated Rag1^-/-^ mice. Interestingly, similar findings were found in WT mice subjected to cerebral ischemia depleted of microglia (Szalay et al., 2016). This suggests that microglia-T cell interaction is required to influence the development of the infarct. We have added this new data set in the new Figure 1e-g of the revised manuscript.

Additionally, we aimed to perform the suggested experiments using T cell-specific IL-10 deficiency (by transfer into lymphocyte-deficient mice). However, as explained above in more detail, we faced considerable delays in animal deliveries from Jackson laboratories to Germany and until today (more than six months after ordering the animals) we still did not receive the transgenic mice. We therefore decided to finalize this revised manuscript without performing this specific experiment.

5. Given the differential involvement of lymphocyte subsets in stroke pathophysiology it is very important to know any impact the lack of lymphocytes may have on factors such as stroke lesion size and infiltration of other immune cells which could skew the microglial profile.

As suggested, we performed additional experiments to address the impact of the lack of lymphocytes on the stroke lesion (new Supplementary Figure 3c and d). Additionally, we performed a comprehensive flow cytometry analysis to also characterize the impact on infiltration of other immune cells (new Supplementary Figure 3a, b, data shown in response 1 above).

Essential experiments:1. Comparison of gene expression between WT and RAG2-/- in naive animals.2. Pathology analysis of stroke lesions in experimental models3. Flow cytometry to determine the numbers of neutrophils, monocyte and macrophages present in the brain in experimental models.4. Care must be taken in text to identify where non-microglial cells may be contributing to effects observed.

We thank the editor for the careful assessment of our manuscript. All concerns raised in the comment and all essential experiments have been addressed during the revision:

We performed the requested gene expression analysis between WT and Rag1^-/-^ naive mice (Figure 2b)We added histological analysis of the stroke lesion (new Suppl. Figure 3c,d)We performed a comprehensive flow cytometry analysis of various leukocyte subpopulations (new Suppl. Figure 3a,b)The text has been thoroughly revised to address the contribution of other resident cell populations and avoid over-interpretation.

Reviewer #1 (Recommendations for the authors):The authors explored the role of T cells in the microglial response to experimental stroke. Using several genetic, cellular, and molecular approaches, they provide evidence that the microglial response can be influenced by T cell signaling. The authors suggest that T cell-related treatment may have a role in preventing the secondary damage that occurs following stroke.Strengths:1. The study employs single-cell sequencing to specifically examine microglial and invading macrophage gene expression profiles following experimental stroke.2. The importance of specific T cell populations was tested using adoptive transfer of Th1 and Treg cells. This approach demonstrates that Th1 T cells can result in upregulation of microglial activation genes, and that Treg cells can result in a more protective gene expression activation pattern.3. The authors used a combination of techniques to identify and verify the gene expression patterns linked to T cell-microglial interactions.Weaknesses:1. Rag-1 deficient mice lack mature T cells and B cells. They are an excellent model to study the role of the immune system in disease models but findings are not specific to T cells.2. The microglial morphological differences shown in Figure 1C are significant but not distinct profiles. In addition, the profile of reactive and homeostatic microglia are not distinguished between WT and Rag-1 deficient mice. Several of the follow up observations are noted as significant but this is not visually evident.3. The adoptive transfer experiments were conducted only in RAG^-/-^ mice so there is not supporting evidence that this T cell-related response would occur in a WT context.4. The rationale for the IL-10 engineered T cells is not provided. The experimental flow does not extend to this experiment, and yet, it is the primary topic of the discussion.5. The reliance on bioinformatic pathway analysis without further direct validation limits the impact of the findings.6. The conclusion that specific T cells drive the polarization of microglia in response to experimental stroke is suggested from the results but not demonstrated.A clear overview of the timeline and schematic of the different experiments is needed.

We have now added a schematic of the timeline for the experiment performed in Figure 4 (see new Figure 4c). Figures 1, 2 and 3 already contain a schematic of the experimental design. For Figure 2 it is mentioned as follow: Lines 143-144: “CD45^+^CD11b^+^ myeloid cells were sorted by flow cytometry from naïve mice or 5 days after stroke (pool of 3 mice per condition) (Figure 1a and Figure 2a).“

It is not clear that the authors integrated the findings across the different experimental approaches.

We have now better integrated findings as follow:

Findings from Figure 1: Lines 119-120: “These data suggest that microglia-T cell interaction is required to influence the development of the infarct.”Findings from Figure 2: Lines 208-211: “Because previous findings showed the CD4^+^ T cell subpopulations exert distinct effects during the post-stroke immune response (Liesz et al., 2009; Gu et al., 2012), we next determined whether functionally different T cell subsets induce or suppress genes in microglia related to cytokine production or cell migration as observed in Figure 2e.”Findings from Figures 2: Lines 237-240 “Interestingly, this set of chemotactic genes induced by T_REG_ cells were also differentially expressed in microglia isolated from Rag1^-/-^ mice in comparison to WT mice (Figure 2b and e), suggesting that microglia can attract different T cell subsets via chemokines, predominantly T_REG_ cells that are known to exert beneficial functions in cerebral ischemia.” And Lines 276-278: “Together, these findings support that polarized T cells, particularly T_REG_ cells, are recruited to the infarction site probably by inducing chemotaxis transcriptional changes in microglia, and may modify in situ the inflammatory micromilieu.”Findings from Figure 3: Lines 281-283: “In order to further explore the implication of T_REG_-microglia interactions to modulate the post-stroke inflammatory environment, we tested the therapeutic potential of the known anti-inflammatory properties of T_REG_ cells via IL10 on the local microglial immune milieu.”

Many genes are included in the narrative but a consensus of the cellular and molecular mechanisms that are participating is not clear.

We fully agree with the reviewer that the simple list of regulated genes is not very informative on the cellular function which might be regulated. However, as requested by another reviewer, we also give in the revised manuscript now a list of the regulated genes (WT versus Rag1^-/-^, see new Figure 2b) but also a more detailed analysis of regulated gene modules (i.e. functional pathways). For example, we find several pathways regulated that are crucial for microglial function in health and disease, including “cytokine/chemokine secretion”, “migration” and “chemotaxis” (see Figures 2 and 3).

Nevertheless, due to the lack of studies systematically correlating transcriptional profiles with specific microglial cell function (i.e. secretion, migration, phagocytosis), the transcriptional information remains descriptive and is not a direct proof of altered cellular function.

Yet, the aim of using transcriptional methods in this study was mainly to quantify the impact of T cells in general (Rag1^-/-^ mice) and of specific T helper cell populations (Treg, Th1) on microglia. Correspondingly, we successfully were able to observe robust transcriptional changes as a proof-of-concept that T cells can shape the microglial response to stroke (on a transcriptional level). The detailed microglial function involves multiple distinct and partially opposing features (such as migration versus proliferation; secretion and phagocytosis) and is spatio-temporarily regulated during the time course after stroke and distance to the lesion border. Therefore, a detailed cellular analysis of all these functions in relation to the influence by T cells is beyond the scope of this proof-of-concept study and will require multiple follow-up studies.

Reviewer #2 (Recommendations for the authors):[…]The following data would strengthen the manuscript and allow a full interpretation of which factors may be influencing microglial phenotype in this manuscript.1. Analysis of differentially expressed genes in naive WT and Rag1-/- microglia to understand baseline effects of the absence of lymphocytes in microglia.

As suggested by the reviewer, we have re-analyzed the transcriptomic data and added a new figure panel to the revised manuscript indicating the differentially expressed genes in naive WT versus Rag1^-/-^ microglia (new Figure 1b). Few genes were significantly regulated in naïve Rag1ko mice in comparison to WT. Interestingly the ones that are up- or down-regulated in naïve condition are oppositely regulated after stroke, suggesting a stroke-dependent effect on microglial gene regulation.

And lines 144-155: “Volcano plot of the differentially expressed genes in naïve condition indicates an upregulation of known microglial genes such as *Apoe* and *Cd74* in CD45^+^CD11b^+^ cells isolated from Rag1^-/-^ mice (Figure 2b, left plot), suggesting that the transcriptional profile of microglia/myeloid cells is already affected in homeostatic condition. However, after inducing stroke these genes were oppositely regulated in Rag1^-/-^ mice in comparison to WT mice together with other genes, including the chemotaxis genes *Ccl2* and *Ccl7* (Figure 2b, right plot, right quadrant: up-regulated in stroke WT) involved in monocyte recruitment but also T cells (Llovera et al., 2017; Popiolek-Barczyk et al., 2020). Together, this indicates an additional effect of stroke in CD45^+^CD11b^+^ cells isolated from Rag1^-/-^ at both the transcriptional and morphological levels (Figure 1, 2). To better discriminate the transcriptional signature of microglial cells from other myeloid cells, we performed an unsupervised clustering analysis and identified 14 distinct clusters across conditions (Figure 2c and Supplementary Figure 1a).”

2. Quantification of infarct sizes in Rag1-/- and WT animals after stroke.

We have now quantified the infarct volumes in the Rag1^-/-^ versus WT (related to Figure 1a-d) and also in Rag1^-/-^ receiving Th1 or Treg cells (Figure 3). No difference in infarct volumes were observed between conditions, suggesting the transcriptional changes in microglia is primarily due to T cell subsets and not due to the primary ischemic lesion. We added this information in the new Supplementary Figure 3 and in the revised manuscript text, lines 272-275: “Importantly, no difference in infarct volumes were observed between WT and Rag1^-/-^ mice (Supplementary Figure 3c) and in T_H1_ or T_REG_-supplemented Rag1^/-^ mice (Supplementary Figure 3d), suggesting the transcriptional changes in microglia is primarily due to T cell subsets and not due to the primary ischemic lesion.”

Of note, the quantification of microglia morphology by histology in Figure 1c, d was performed in 3 mice with similar infarct volume to exclude bias by variability in the primary lesion severity

3. Flow cytometry analysis of immune cell populations present in the brain at 5d post stroke to determine the number and proportion of neutrophils, monocytes and macrophages in RAG^-/-^ and WT animals.

As requested by the reviewer, we performed an additional experiment and analyzed various additional leukocyte subpopulations in the ipsilesional hemisphere 5 days after dMCAO in wild-type (WT) and Rag-1 deficient mice by flow cytometry. This additional experiment revealed that cell counts for innate immune cell populations were not affected by the Rag1 gene deletion, supporting the conclusion that lymphocytes, and most likely T cells are the main contributor to the observed microglial phenotype.

The detail of these results can be found in the new Supplementary Figure 3a, b and lines 201208: “Because Rag1^-/-^ mice lack mature T cells and B cells, it is possible that the observed morphological and transcriptional changes of microglia may be due to B cells or other myeloid cell types. We performed flow cytometry analysis of the ipsilesional hemisphere 5 days after dMCAO in WT and Rag1^-/-^ mice. First, we demonstrate that T cells are 14 times more abundant than B cells in the ipsilesional hemisphere in WT mice (Supplementary Figure 3a). In addition, the abundance of myeloid cell subsets is not affected by the Rag1 gene deletion (Supplementary Figure 3b). These data support the hypothesis that lymphocytes and most likely T cells are the main contributor to the observed microglial phenotype at this time point after stroke.”

4. Image analysis treating an individual microglia as a data point is pseudoreplication. This should be averaged within one animal, which is the experimental unit in this design, and analysed appropriately.

We do understand the reviewer’s concern, however, we have to disagree on the statement that the individual mouse is the experimental unit for biological replicates to be analyzed. With this rationale stated by the reviewer also data from single-cell sequencing, multiphoton imaging of single cells or analysis of blood flow at individual blood vessels would be needed to be analyzed at the level of individual mice – which is not meaningful and does also not represents current scientific practice. In fact, this would be the equivalent to requesting single-cell sequencing data to be analyzed as bulk data per mouse. Similar to the examples given above, also automated morphological analysis of microglia is an analysis strategy which provides multiple parameters (64 shape parameters are analyzed by the analysis tool) per single cell. That way each individual cell is treated as an independent biological replicate characterized by a multiparametric descriptor; the individual mouse represents individual experiments. Considering the variability of microglial reaction states – that are reflected in the morphological changes – simply merging all single-cell morphology data in a mean per mouse would ignore the cell-to-cell variability, artificially simplify the high dimensionality of the data and ignore the differences between microglial subpopulations. Moreover, this way of data analysis of automated microglia morphology analysis is well established in the literature and used in numerous publications, including previous studies published in *eLife* (e.g. Colombo et al., *eLife*, 2021).

Moreover, analysis of individual cells cannot be regarded as (uncontrolled) pseudo-replication because the used statistical test is correcting for this. Attribution of cells to individual mice (i.e. individual experiments) has been used as an independent factor for posthoc analysis – as would be used in other single-cell analyses to test for batch effects which might distort the data structure. Importantly, we did not find a significant effect of group allocation by individual mouse for any of the microglia morphology data sets in this manuscript.

5. The n number of animals going into experiments is not always clear in figure legends.

We have added this information in each of the figure legends.

In the discussion, it should be acknowledged that T cells are not the only immune cell that may influence microglial polarisation in this manner. IL-10 producing regulatory B cells should be discussed.

We fully agree with the reviewer that IL-10 producing Breg cells might also contribute to a microglia-modulating effect; accordingly, we have added a statement on Breg cells to the Discussion section:

Lines 353-356: “However, we cannot exclude in this study the contribution of IL-10 from other lymphocyte subpopulations, particularly IL-10-producing regulatory B cells (Bodhankar et al., 2013; Ortega et al., 2020; Seifert et al., 2018), as we did not specifically deplete IL-10 only in T cells.”

The authors should justify why no sham-operated controls were used in these experiments.

In this study we did not focus on the impact of stroke on the pathology, and specifically microglia, *per se* but the role of T cells in modulating the microglial response after stroke. T cells do not invade the brain in sham condition nor can we observe overt microglial activation after sham surgery – in accordance with numerous studies analyzing the neuroinflammatory response between stroke and sham, including multiple studies from our laboratory and others (Liesz et al., 2011; Llovera et al., 2015). Therefore, most of the key findings on the functional role of T cells on modulating outcome were performed in stroke mice (+/- T cells) and naïve mice were only used for transcriptomic studies to enrich for homeostatic cells in comparison to stroke-induced activated microglia.

The authors should discuss the use of RNA extracted from whole brain hemisphere in the final experiment and highlight that cells that are not microglia may contribute to gene expression changes identified.

It is correct that the RNA profile in Figure 4 could be contaminated by other cell types than microglia. We have now stated this aspect more clearly: Lines 302-305: “Although the observed regulated genes are well known to be associated with microglial function, it is conceivable that in this analysis other cell types than microglia, including various brain-invading myeloid cell subsets, could account for this effect since the whole ischemic brain tissue was processed for Nanostring analysis.”

For technical reason we could not sort microglia and quantify the infarct lesion from the same brain. Because we wanted to address here the therapeutic potential of the engineered Tc-IL10 in inducing neuroprotection, we collected sequential coronal sections from frozen brain for either histological analysis of the lesion volume and for transcriptomic analysis from the whole ipsilesional area which prevented us to perform single cell sorting of microglia.

Reviewer #3 (Recommendations for the authors):[…]1. Related to Figures 1c, 3e-f, 4g: It is not appropriate to use individual microglia as n values when quantifying microglia morphology, RNAscope puncta in microglia, etc. Instead, mice should be used as biologically independent replicates. It does not appear that Trem2 expression differences in Figure 3e and 4g would be anywhere near statistically significant given the variation if appropriate statistics were applied. Similarly, we have concerns about the statistics applied to the SmartSeq2 data presented in Figure 2f. The authors should use a differential expression method designed specifically for single cell sequencing data to compare WT to KO microglia. Again, using individual cell pools as n values in this case when running statistical tests massively inflates the resulting p value and makes very small changes in gene expression erroneously appear significant.

Regarding the microglia morphometry analysis:

Automated microglial morphometry is a high-dimensionality, cell-based analysis method. It was initially described and established by now as widely used tool to describe 3-dimensional morphology of individual cells. This is also reflected in numerous published studies across different labs and disease models (e.g. Sadler, J Neurosci, 2020; Cserep, Science, 2020; Guldner, Cell, 2020; Otxoa-de-Amezaga, Acta neuropathologica, 2019; Chen, PNAS, 2019; Fernandez-Arjona, Front Cell Neurosci, 2019; etc.) which used this tool and analyzed microglia morphology by using the single cell as a biological unit (instead of the whole organism), including several publications with the same tool and analysis pipeline published in *eLife* (e.g. (Colombo et al., 2021; Wildenberg et al., 2021)).

Segmentation of single cells from the 3D-recontructed imaging data set occurs very early in the analysis pipeline, so that all shape descriptors (64 in total) are based already on individual cells. The strength of this approach is that it incorporates the large heterogeneity in microglial responses to stimuli which is reflected in their morphology. Likewise, we see clusters of cells with a different morphological response in comparison to cells that are more distant to the lesion or have received different signals. Hence, the analysis of this multidimensional cell-based information is in analogy to the transcriptional information on single cells obtained from single-cell sequencing methods. Therefore, analyzing the obtained data with means on organism (mouse) as the biological unit would be comparable as requesting to analyze singlecell sequencing data as bulk transcriptomic information per animal.

Since all of the analysis is done per single cells, it is in our opinion not only justified to perform also the bioinformatic analysis using cells as biological replicates (and using mice as individual experiments) but it also better reflects the large variability in cell morphology within the heterogenous population – information which would be completely lost by the bulk analysis.

In order to exclude bias and distortion in data distribution of single cells by individual mice, we performed a linear mixed model analysis using mouse as grouping variable which revealed no significant effect (P=0.39) for grouping by mice. This has also been corrected for by posthoc test for the presented data in the revised manuscript.

Regarding the Smart Seq2 transcriptomic analysis:

According to the reviewer’s suggestion, we used the DEseq method (Anders and Huber, 2010) that is suitable for non UMI single cell sequencing data. As expected DEseq analysis revealed a clear stroke effect on microglial transcriptome and nearly no effect significant effect of RAG-KO which indicates our statistical method sensitive enough to capture the stoke phenotype but did not make very small changes in gene expression erroneously appear significant and found genotype effect (see Author response image 1).

**Author response image 1. sa2fig1:** 

However, since the Smart Seq2 analysis does not provide much additional information to the revised manuscript in comparison to the revised and improved 10x single-cell analyses and other new experiments and analyses provided during the revision, we decided to remove this data set from the revised manuscript.References

Anders S. and Huber W., (2010). Differential expression analysis for sequence count data. Genome Biol 11, R106–R106. https://doi.org/10.1186/gb-2010-11-10-r106

Andres R.H., Choi R., Pendharkar A.V., Gaeta X., Wang N., Nathan J.K., Chua J.Y., Lee S.W., Palmer T.D., Steinberg G.K., and Guzman R., (2011). The CCR2/CCL2 Interaction Mediates the Transendothelial Recruitment of Intravascularly Delivered Neural Stem Cells to the Ischemic Brain. Stroke 42, 2923– 2931. https://doi.org/10.1161/strokeaha.110.606368

Berchtold D., Priller J., Meisel C., and Meisel A., (2020). Interaction of microglia with infiltrating immune cells in the different phases of stroke. Brain Pathol 30, 1208–1218. https://doi.org/10.1111/bpa.12911

Bodhankar S., Chen Y., Vandenbark A.A., Murphy S.J., and Offner H., (2013). IL-10-producing B-cells limit CNS inflammation and infarct volume in experimental stroke. Metab Brain Dis 28, 375–86. https://doi.org/10.1007/s11011-013-9413-3

Coenye T., (2021). Do results obtained with RNA-sequencing require independent verification? Biofilm 3, 100043. https://doi.org/10.1016/j.bioflm.2021.100043

Colombo A.V., Sadler R.K., Llovera G., Singh V., Roth S., Heindl S., Monasor L.S., Verhoeven A., Peters F., Parhizkar S., Kamp F., Aguero M.G. de, MacPherson A.J., Winkler E., Herms J., Benakis C., Dichgans M., Steiner H., Giera M., Haass C., Tahirovic S., and Liesz A., (2021). Microbiota-derived short chain fatty acids modulate microglia and promote Aβ plaque deposition. *ELife* 10, e59826. https://doi.org/10.7554/*eLife*.59826

Dimitrijevic O.B., Stamatovic S.M., Keep R.F., and Andjelkovic A.V., (2007). Absence of the Chemokine Receptor CCR2 Protects Against Cerebral Ischemia/Reperfusion Injury in Mice. Stroke 38, 1345–1353. https://doi.org/10.1161/01.str.0000259709.16654.8f

Gelderblom M., Leypoldt F., Steinbach K., Behrens D., Choe C.-U., Siler D.A., Arumugam T.V., Orthey E., Gerloff C., Tolosa E., and Magnus T., (2009). Temporal and Spatial Dynamics of Cerebral Immune Cell Accumulation in Stroke. Stroke 40, 1849–1857. https://doi.org/10.1161/strokeaha.108.534503

Hammond T.R., Dufort C., Dissing-Olesen L., Giera S., Young A., Wysoker A., Walker A.J., Gergits F., Segel M., Nemesh J., Marsh S.E., Saunders A., Macosko E., Ginhoux F., Chen J., Franklin R.J.M., Piao X., McCarroll S.A., and Stevens B., (2019). Single-Cell RNA Sequencing of Microglia throughout the Mouse Lifespan and in the Injured Brain Reveals Complex Cell-State Changes. Immunity 50, 253271.e6. https://doi.org/10.1016/j.immuni.2018.11.004

Keren-Shaul H., Spinrad A., Weiner A., Matcovitch-Natan O., Dvir-Szternfeld R., Ulland T.K., David E., Baruch K., Lara-Astaiso D., Toth B., Itzkovitz S., Colonna M., Schwartz M., and Amit I., (2017). A Unique Microglia Type Associated with Restricting Development of Alzheimer’s Disease. Cell 169, 1276-1290.e17. https://doi.org/10.1016/j.cell.2017.05.018

Kleinschnitz C., Schwab N., Kraft P., Hagedorn I., Dreykluft A., Schwarz T., Austinat M., Nieswandt B., Wiendl H., and Stoll G., (2010). Early detrimental T-cell effects in experimental cerebral ischemia are neither related to adaptive immunity nor thrombus formation. Blood 115, 3835–3842. https://doi.org/10.1182/blood-2009-10-249078

Li L., Lou W., Li H., Zhu Y., and Huang X., (2020). Upregulated C-C Motif Chemokine Ligand 2 Promotes Ischemic Stroke via Chemokine Signaling Pathway. Ann Vasc Surg 68, 476–486. https://doi.org/10.1016/j.avsg.2020.04.047

Liesz A., Zhou W., Mracskó É., Karcher S., Bauer H., Schwarting S., Sun L., Bruder D., Stegemann S., Cerwenka A., Sommer C., Dalpke A.H., and Veltkamp R., (2011). Inhibition of lymphocyte trafficking shields the brain against deleterious neuroinflammation after stroke. Brain 134, 704–720. https://doi.org/10.1093/brain/awr008

Llovera G., Benakis C., Enzmann G., Cai R., Arzberger T., Ghasemigharagoz A., Mao X., Malik R., Lazarevic I., Liebscher S., Ertürk A., Meissner L., Vivien D., Haffner C., Plesnila N., Montaner J., Engelhardt B., and Liesz A., (2017). The choroid plexus is a key cerebral invasion route for T cells after stroke. Acta Neuropathol 134, 851–868. https://doi.org/10.1007/s00401-017-1758-y

Llovera G., Hofmann K., Roth S., Salas-Pérdomo A., Ferrer-Ferrer M., Perego C., Zanier E.R., Mamrak U., Rex A., Party H., Agin V., Fauchon C., Orset C., Haelewyn B., Simoni M.-G.D., Dirnagl U., Grittner U., Planas A.M., Plesnila N., Vivien D., and Liesz A., (2015). Results of a preclinical randomized controlled multicenter trial (pRCT): Anti-CD49d treatment for acute brain ischemia. Sci Transl Med 7, 299ra121299ra121. https://doi.org/10.1126/scitranslmed.aaa9853

Llovera G., Roth S., Plesnila N., Veltkamp R., and Liesz A., (2014). Modeling Stroke in Mice: Permanent Coagulation of the Distal Middle Cerebral Artery. J Vis Exp Jove 51729. https://doi.org/10.3791/51729

Ortega S.B., Torres V.O., Latchney S.E., Whoolery C.W., Noorbhai I.Z., Poinsatte K., Selvaraj U.M., Benson M.A., Meeuwissen A.J.M., Plautz E.J., Kong X., Ramirez D.M., Ajay A.D., Meeks J.P., Goldberg M.P., Monson N.L., Eisch A.J., and Stowe A.M., (2020). B cells migrate into remote brain areas and support neurogenesis and functional recovery after focal stroke in mice. Proc National Acad Sci 117, 4983– 4993. https://doi.org/10.1073/pnas.1913292117

Popiolek-Barczyk K., Ciechanowska A., Ciapała K., Pawlik K., Oggioni M., Mercurio D., Simoni M.-G.D., and Mika J., (2020). The CCL2/CCL7/CCL12/CCR2 pathway is substantially and persistently upregulated in mice after traumatic brain injury, and CCL2 modulates the complement system in microglia. Mol Cell Probe 54, 101671. https://doi.org/10.1016/j.mcp.2020.101671

Safaiyan S., Besson-Girard S., Kaya T., Cantuti-Castelvetri L., Liu L., Ji H., Schifferer M., Gouna G., Usifo F., Kannaiyan N., Fitzner D., Xiang X., Rossner M.J., Brendel M., Gokce O., and Simons M., (2021). White matter aging drives microglial diversity. Neuron 109, 1100-1117.e10. https://doi.org/10.1016/j.neuron.2021.01.027

Seifert H.A., Vandenbark A.A., and Offner H., (2018). Regulatory B cells in experimental stroke. Immunology 154, 169–177. https://doi.org/10.1111/imm.12887

Szalay G., Martinecz B., Lénárt N., Környei Z., Orsolits B., Judák L., Császár E., Fekete R., West B.L., Katona G., Rózsa B., and Dénes Á., (2016). Microglia protect against brain injury and their selective elimination dysregulates neuronal network activity after stroke. Nat Commun 7, 11499. https://doi.org/10.1038/ncomms11499

Wildenberg G., Sorokina A., Koranda J., Monical A., Heer C., Sheffield M., Zhuang X., McGehee D., and Kasthuri B., (2021). Partial connectomes of labeled dopaminergic circuits reveal non-synaptic communication and axonal remodeling after exposure to cocaine. *ELife* 10, e71981. https://doi.org/10.7554/*eLife*.71981

[Editors’ note: further revisions were suggested prior to acceptance, as described below.]

The manuscript has been improved but there are some remaining issues that need to be addressed. As you revise your manuscript, please take note of the comments below, in particular, those from Rev. 1Reviewer #1 (Recommendations for the authors):The authors have attempted to address previous comments and have done so in some cases. There are still several important controls missing that would allow them to make the link between T cells and microglia phenotype/function in stroke.Addition of functional evidence to support this link is needed to support the claims – otherwise showing how T cells polarize microglia transcriptionally after stroke doesn't seem like a substantial advance in the field. Also, the fact that deleting microglia makes the infarct worse is confusing and makes it really difficult to understand how T cells program microglia to cause worse pathology later. It is likely that microglia play both protective and damaging roles, but they would need more elegant experiments or manipulation of specific pathways in microglia to tease that apart.

We thank Reviewer 1 for the thorough evaluation of our revised manuscript. All remaining concerns have been addressed below and in the revised manuscript.

Specific Comments:1. The authors now show changes between WT and Rag2KO myeloid cells after stroke as individual genes in addition to pathways. This addresses our comment. However, since this is single cell sequencing data, the authors should make DE gene expression comparisons within individual cell types (e.g. WT vs KO microglia) instead of pooling data across cell types.

In our revised manuscript, we have indeed presented the volcano plots of the differentially expressed genes from the total CD45+CD11b+ population and agree with the Reviewer that presenting the DEG from the microglia clusters only would be more specific. Accordingly, we have now modified the analysis to include only cells identified as microglia as presented in Figure 1D. Results for DEG analysis from this cluster is now presented in the revised Figure 2. We have also added a new analysis in Figure 2e, highlighting the stroke-associated microglial genes that are specifically regulated in microglia from the lymphocyte deficient mice, showing clearly that genes associated with cytokine signaling and chemotaxis are regulated in microglia after stroke in the absence of lymphocyte. The text in the Results section has been modified accordingly.

2. We appreciate the addition of the microglia depletion experiment in figure 1g. However, this experiment still does not address the question of whether T cell-microglia interaction affects stroke pathology because it is missing the WT control and depletion conditions. From the data presented, all we can conclude is that microglia influence stroke pathology on a Rag2KO background. Depletion of microglia may well have the same effect in WT mice, which would argue against a role for T cell priming of microglia function.

As suggested by the reviewer, we have now added a WT control group and the group for microglia depletion in WT mice. Comparing infarct volumes between these four groups, we observed a significant increase in infarct volume after microglia depletion only in lymphocytedeficient but not in WT mice, suggesting that depletion of microglia in WT mice does not have the same effect as in the lymphocyte deficient mice.

3. In general, the authors need to revise their discussion of the role of T cells in influencing microglia "function" after stroke. Without clear experiments linking T cells -> microglia -> pathology, or alternatively T cells to modulation of a functional readout in microglia (e.g. engulfment), the authors can only conclude that T cells modulate microglia morphology and transcriptome as they have shown throughout the manuscript.e.g. line 340: "Here, we established a mechanistic link between T cells and microglial function and showed the distinct role of T cell subpopulations on switching microglial polarization state in response to stroke."

It is correct that so far we did not directly test whether T cells modulate functional changes of microglia but rather microglial morphology and transcriptomic profile. In order to avoid any misinterpretation or overstatement, we have now modified the summary and discussion as follow:

“The crosstalk between brain infiltrating T cells and microglia in response to stroke remains elusive. Benakis et al. report that transcriptional signature of the stroke-associated microglia is reprogrammed by distinct T cell subpopulations. Engineered T cells overexpressing IL-10 administered four hours after stroke reinitiate microglial transcriptomic profile inducing a pro-regenerative environment.”

“Here, we established a mechanistic link between T cells and microglial morphology and transcriptomic signature in the context of stroke. We showed the distinct role of T cell subpopulations on switching microglial polarization state in response to stroke.”

“We demonstrated that IL-10 overexpression by this approach substantially modulated microglia gene expression by down-regulation of microglial gene signature associated with phagocytosis of synapses correlating with functional recovery after stroke.”

“Whereas at this acute time point the transcriptomic changes in microglia are mainly attributed to their reactivity to the tissue injury itself, we have been able to demonstrate that brain-invading T cells can specifically “fine-tune” the transition of the stroke-associated microglia to a distinct cell morphology and transcriptomic profile. Our data suggested that the anti-inflammatory T_REG_ cells induce a shift of microglial genes associated with a homeostatic state and immune cell recruitment. However, the specific functional change of microglia induced by T cell subsets and biological significance for stroke remain to be further investigated.”

Reviewer #2 (Recommendations for the authors):The authors have addressed the comments and conducted additional experiments that have improved the manuscript. The edited results narrative provides a better guide to the key findings and the additional results allow better interpretation of the original findings.It would be good to update the Summary and Abstract to highlight key take homes. In particular, the impact of the stroke itself on lesion size, as well as the particular potential protective or detrimental role for specific T cells.

We thank the Reviewer for the positive feedback on our revised manuscript. As suggested we have now modified the abstract as follow:

“Here, using a mouse model for ischemic stroke, we demonstrated that early activation of microglia in response to stroke is differentially regulated by distinct T cell subpopulations – with T_H1_ cells inducing a type I interferon signaling in microglia and T_REG_ cells promoting microglial genes associated with chemotaxis. Acute treatment with engineered T cells overexpressing IL-10 administered into the cisterna magna after stroke induces a switch of microglial gene expression to a profile associated with proregenerative functions. Whereas microglia polarization by T cell subsets did not affect the acute development of the infarct volume, these findings substantiate the role of T cells in stroke by polarizing the microglial phenotype. Targeting T cell-microglia interactions can have direct translational relevance for further development of immune-targeted therapies for stroke and other neuroinflammatory conditions.”

Reviewer #3 (Recommendations for the authors):The authors have addressed the majority of concerns with the manuscript and the new analyses included allow for a more transparent interpretation of the data within. Gene expression changes between naive WT and Rag1-/- mice, flow cytometry of innate immune cells in WT and Rag1-/- after stroke and information on stroke lesion size all help to exclude non-T cell related factors that may influence microglial gene expression. The linear mixed model analysis, with grouping as a variable, improves the statistical analysis performed on microglial morphology data.Furthermore, the alterations to the text have reduced over interpretation of the data in the manuscript and better reflect the data shown.Although the authors were unable to perform an experiment using IL-10R -/- microglia due to difficulties in receiving the appropriate mouse lines, the final experiment in the manuscript shows that transfer of IL-10 over expressing T cells had a biological effect on microglia in the context of a conventional neuroinflammatory response to stroke in WT mice. Therefore although direct signalling between the IL-10+ T cells and the microglia has not been definitively proven, the resulting effects on microglial gene expression and stroke functional outcome show this to be a promising therapeutic strategy.I believe the authors have satisfied the reviewers original comments with no further changes required.

We agree that the direct signaling between T cells secreting IL-10 and the impact on microglia remains to be further investigated. We added this aspect in the discussion as indicated in response to Reviewer 1. We thank Reviewer 3 for the constructive comments on our manuscript.